# Lactobacilli-Derived Microbe-Associated Molecular Patterns (MAMPs) in Host Immune Modulation

**DOI:** 10.3390/biom15111609

**Published:** 2025-11-17

**Authors:** Salvatore Furnari, Ruben Ciantia, Adriana Garozzo, Pio Maria Furneri, Virginia Fuochi

**Affiliations:** Department of Biomedical and Biotechnological Sciences (BIOMETEC), University of Catania, 95123 Catania, Italy; salvatore.furnari@phd.unict.it (S.F.); agar@unict.it (A.G.); pio.furneri@unict.it (P.M.F.)

**Keywords:** host–microbe interactions, microorganism-associated molecular patterns (MAMPs), Toll-like receptors (TLRs), immunomodulation, immune response, inflammatory response, pattern recognition receptors (PRRs)

## Abstract

Although traditionally sidelined by live probiotic effects, Lactobacilli-derived Microbe-Associated Molecular Patterns (MAMPs) are emerging as potent modulators of innate and adaptive immune responses, capable of acting independently of bacterial viability. However, the underlying mechanisms remain incompletely understood. These MAMPs, such as peptidoglycan (PGN), lipoteichoic acid (LTA), and exopolysaccharides (EPSs), interact with pattern recognition receptors (PRRs) like Toll-like receptors (TLRs), initiating immune-signaling cascades that regulate cytokine production and inflammation. Lactobacilli-derived MAMPs exhibit dual immunomodulatory effects: they can enhance pro-inflammatory responses, e.g., interleukin-1β (IL-1β), IL-6, and tumor necrosis factor alpha (TNF-α) under inflammatory contexts, while enhancing regulatory pathways via IL-10 and regulatory T-cell (T_regs_) induction in anti-inflammatory settings. Importantly, these immunomodulatory properties persist in the absence of bacterial viability, making MAMPs promising candidates for postbiotic therapies. This opens new avenues for MAMP-based strategies to target inflammation, overcoming the risks associated with live bacterial administration. This review examines the therapeutic relevance of non-viable MAMPs, particularly in inflammatory diseases where they have demonstrated benefits in reducing tissue damage, enhancing gut barrier function, and alleviating disease symptoms. Additionally, we discuss regulatory and translational challenges hindering their clinical implementation, highlighting the need for standardized characterization, a clear safety framework, and strain-specific profiling. Given their ability to fine-tune immune responses, MAMPs represent an emerging strategy for innovative treatments aimed at restoring immune balance and reinforcing host–microbe interactions.

## 1. Introduction

The family *Lactobacillaceae* (for simplicity, the term “lactobacilli” will hereafter be used to collectively refer to all members of this family) comprises Gram-positive, non-spore-forming, facultatively anaerobic bacteria belonging to the phylum Firmicutes, currently encompassing over 260 species across 25 genera [1,2]. Lactobacilli constitute a taxonomically and phylogenetically diverse group of bacteria that colonizes multiple mucosal surfaces, including gastrointestinal, oral, and genitourinary tracts. At these interfaces, lactobacilli contribute to mucosal immune education and microbial homeostasis through mechanisms that go far beyond passive colonization, influencing both immune tolerance and pathogen resistance [3,4,5].

Beyond the direct suppression of pathogens, lactobacilli exert a broad spectrum of immunological effects that influence innate and adaptive responses, contributing to mucosal homeostasis and reducing the risk of chronic inflammatory diseases [3,6]. These effects are mediated, in part, by the production of antimicrobial and signaling molecules such as bacteriocins, reactive oxygen species (ROS), and organic acids that shape both microbial composition and host responses [7,8,9,10]. Furthermore, lactobacilli enhance intestinal epithelial barrier integrity, engage in the competitive exclusion of pathogens, and exert oncopreventive and antiproliferative effects, establishing them as valuable components of probiotic formulations [1,11,12]. These functional attributes justify their widespread use as probiotics, defined by the International Scientific Association for Probiotics and Prebiotics (ISAPP) as “live microorganisms that, when administered in adequate amounts, confer health benefits on the host” [13]. Supported by their Generally Recognized as Safe (GRAS) status [14] and decades of clinical applications, lactobacilli are today incorporated into a variety of probiotic formulations, including foods containing lactobacilli and their metabolites, such as yogurt. Notably, evidence suggests that their beneficial effects may extend beyond the gut, including effects on extra-intestinal carcinogenesis, underscoring their potential role in cancer prevention and overall health improvement [14,15,16,17].

While these effects have traditionally been attributed to the activity of live probiotic bacteria, several studies now demonstrate that inactivated cells, structural components, and secreted molecules can exert comparable functions [3,18,19]. This has led to the ISAPP definition of postbiotics as “preparations of inanimate microorganisms and/or their components that confer a health benefit on the host” [20]. Among these components, MAMPs have been identified as biologically active mediators that may account for many of the immune-regulatory properties previously ascribed to viable lactobacilli [21].

A key element of their immunoregulatory capacity is the production of MAMPs, detected by host PRRs, including TLRs and NOD-like receptors (NLRs). As summarized in Appendix A, key lactobacilli-derived MAMPs include lipoteichoic acid (LTA), peptidoglycan (PGN), and exopolysaccharides (EPSs). These molecules vary structurally among strains, yet converge functionally in modulating immune signaling pathways [22,23]. The recognition of these components by TLR2 and NOD2 has been shown to activate downstream signaling cascades primarily through NF-κB and Mitogen-Activated Protein Kinase (MAPK) pathways, leading to an increased secretion of anti-inflammatory cytokines like IL-10, while reducing the levels of pro-inflammatory mediators including TNF-α and IL-6 [24,25]. Moreover, these signals modulate the activity of dendritic cells (DCs) and other antigen-presenting cells (APCs), resulting in an enhanced induction and differentiation of T_regs_, a mechanism essential for maintaining mucosal immune tolerance and preventing autoimmune responses [26].

This review examines how lactobacilli-derived MAMPs regulate host immunity by engaging PRRs and initiating signaling cascades that affect both innate and adaptive responses. The molecular interactions between these bacterial components and host pattern recognition receptors are described, with a particular focus on the downstream signaling pathways involved in cytokine modulation, T_reg_ induction, and T helper (T_h_) cells. Furthermore, the therapeutic potential of lactobacilli-derived MAMPs in clinical settings is discussed, highlighting their potential as non-viable immunomodulatory agents and addressing current limitations to clinical application. The aim is to clarify the mechanisms by which MAMPs contribute to host immune regulation and to provide a rationale for their integration into postbiotic-based strategies for immune-mediated diseases.

## 2. Materials and Methods

This narrative review was conducted through a comprehensive literature search, primarily in MEDLINE, guided by the review’s focus on immunomodulation. No publication date restrictions were applied, ensuring a broad selection of studies. However, references were included based on their relevance to lactobacilli-derived MAMPs and their interactions with the host immune system.

The main search terms included “*Lactobacillus* MAMPs,” “innate immunity,” “adaptive immunity,” “cytokine modulation,” “regulatory T cells,” “pattern recognition receptors,” “T_h_1/T_h_2 differentiation,” “probiotic properties,” “GRAS status,” “inflammatory bowel disease,” and “autoimmune diseases.” Studies were selected if they provided a detailed exploration of immune modulation mechanisms. Additionally, given the importance of lactobacilli probiotics in immunology, both clinical and preclinical studies on their immunological impact were included.

The focus on cytokines and immune pathways was restricted to those that act as key regulators of inflammation and immune tolerance, ensuring the review concentrated on anti-inflammatory mechanisms and immune regulation. Furthermore, clinically relevant studies were included to provide a comprehensive overview of the therapeutic potential of lactobacilli-derived MAMPs in immune-mediated diseases.

A total of 587 records were initially identified, of which 300 met the inclusion criteria and were included.

## 3. Lactobacilli and the Art of Immune Modulation

The hypothesis that microbial populations could influence host physiology and longevity was first formulated in the early 20th century. Élie Metchnikoff proposed that lactic acid bacteria derived from fermented dairy products contributed to health maintenance by suppressing the proteolytic bacteria responsible for intestinal putrefaction, thus prolonging life expectancy [27]. This early concept, rooted in the demographic observation of Eastern European rural populations, provided the foundational rationale for subsequent probiotic research.

In the decades that followed, the immunological relevance of commensal and probiotic microorganisms, particularly lactobacilli, was increasingly recognized. As shown in Figure 1, these bacteria were shown to interact with the host immune system, influencing both mucosal and systemic responses. Their effects were described as highly strain-dependent and conditional, varying according to the physiological and immunological state of the host [28]. As a result, lactobacilli became central subjects in the research on host–microbe interactions, particularly in the context of immune regulation and chronic inflammatory diseases.

### 3.1. Immune Effects of Live Probiotic Bacteria

#### 3.1.1. Pro-Inflammatory Effects Under Resting Conditions

Under non-inflammatory or resting conditions, several strains of lactobacilli were reported to stimulate the production of pro-inflammatory mediators. This activity was attributed to the induction of nitric oxide synthase (iNOS) and cyclooxygenase-2 (COX2), leading to elevated levels of nitric oxide (NO) and prostaglandin E2 (PGE_2_) [29,30]. Moreover, an increased secretion of pro-inflammatory cytokines, including IL-1α, IL-1β, IL-6, interferon-γ (IFN-γ), and TNF-α, was observed for both in vitro [29,30,31,32,33] and in vivo [34,35] models following lactobacilli administration.

In human intestinal epithelial Caco-2 cells under basal conditions, certain lactobacilli were shown to increase IL-8 expression, suggesting that, even in the absence of exogenous stimuli, lactobacilli can initiate a mild pro-inflammatory response [9]. These observations supported the notion that immunostimulatory effects were not exclusive to pathological contexts and might contribute to basal immune response and mucosal vigilance.

#### 3.1.2. Anti-Inflammatory Response in Inflamed States

When administrated under inflammatory conditions, lactobacilli demonstrated immunosuppressive and anti-inflammatory properties. In epithelial models pre-treated with pro-inflammatory stimuli such as IL-1β or lipopolysaccharide (LPS), the specific strain reduced the secretion of IL-8 and NO production, suggesting a shift towards immune suppression [9].

Similar effects were confirmed in vivo. In endotoxin-challenged murine models, the administration of *Limosilactobacillus reuteri* resulted in a significant decrease in IL-1β and IL-6, along with a concomitant increase in IL-10. Additionally, a reduction in lipid peroxidant activity was observed, suggesting antioxidant activity and the restoration of redox balance in the gut [36].

In this context, the dominance of anti-inflammatory mechanisms appeared to be dependent on the inflammatory status of the host. It was reported that the immunosuppressive capacity of lactobacilli became particularly pronounced after immune activation, leading to the downregulation of pro-inflammatory cytokines and the upregulation of regulatory mediators such as IL-10 and TGF-β [37,38,39,40].

#### 3.1.3. Administration of Live Lactobacilli

Beyond their effects on cytokine profiles, lactobacilli were also shown to exert a wide range of regulatory functions on both innate and adaptive immune cells. In models of enteric infections, such as *Salmonella*-infected mice, the administration of *Lacticaseibacillus rhamnosus* resulted in an enhanced neutrophil activity, increased secretion of inflammatory cytokines, and elevated levels of pathogen-specific antibodies, ultimately reducing the intestinal bacterial burden [34].

In rIFN-γ-primed macrophages, *Latilactobacillus sakei* stimulated NO production and inflammatory cytokines, while also increasing phagocytic activity, indicating that primed innate cells retained a responsiveness to lactobacilli signals [41]. On the adaptive side, lactobacilli promoted T-cell proliferation and differentiation [29,32,42], with various strains shown to modulate T_h_1/T_h_2 balance through a shift in the T-bet/GATA-3 transcription factor ratio [35,43]. Moreover, T_reg_ populations (CD25^+^ Foxp3^+^) were expanded following lactobacilli exposure [32], and NK cell cytotoxicity was enhanced in multiple experimental systems [31].

Additional studies revealed that lactobacilli-conditioned monocytes reduced IFN-γ and IL-17 production by *Staphylococcus aureus*-stimulated T-cells and NK cells via soluble immunoregulatory mediators [44].

In preclinical models of intestinal inflammation, lactobacilli exerted protective effects on epithelial integrity and tissue morphology. Specifically, treatment with selected strains resulted in an upregulation of tight junction proteins such as mucin and occludin, supporting epithelial barrier function in inflamed colonic tissue [38]. In murine models of colitis, various lactobacilli strains significantly attenuated inflammation-induced weight loss [35,38,39], reduced colon length shortening [38,39], and also reduced histopathological alterations, including crypt loss and villus atrophy [35,38]. These have improved the disease activity index (DAI) score [38,39], which assesses body weight, stool consistency, and rectal bleeding [45].

Beyond local effects, beneficial outcomes were also observed systemically. Lactobacilli modulated neuroinflammation in endotoxin-induced models via the gut–brain axis, suggesting a role in the bidirectional communication between intestinal immunity and central nervous system homeostasis [36]. Similarly, the application of *L. salivarius* and *L. rhamnosus* strains led to a reduction in skin inflammation in mice, pointing to the relevance of probiotic action beyond the gastrointestinal tract [46]. Of particular note, *L. rhamnosus* GG (LGG) was shown to activate cytotoxic T-cells, expand tumor-infiltrating dendritic cells (DCs), and promote IFN production, highlighting its immunotherapeutic potential in a cancer model [42]. Furthermore, in immunocompromised hosts such as cyclophosphamide-treated mice, the administration of *Lactiplantibacillus plantarum* restored lymphocyte and antibody levels (IgM, IgA, and IgG) and normalized the granulocyte-to-lymphocyte ratio, which was elevated in untreated animals, suggesting immune reconstitution effects [43]. These findings positioned lactobacilli not only as anti-inflammatory agents but also as modulators of systemic immune responses in diverse disease settings.

### 3.2. Immunomodulatory Effects by Non-Viable Forms

Several studies demonstrated that non-viable lactobacilli, including heat-killed bacteria [30,33,39], cell-free supernatant (CFS) [39,40], and bacterial lysate [47,48], retained significant immunomodulatory functions. These effects suggested that viability was not a prerequisite for immune activation and, instead, pointed towards specific bacterial components as the bioactive agents.

This understanding supported the evolving concept of postbiotics, defined as “preparations of inanimate microorganisms and/or their components that confer a health benefit to the host” [20]. Among the principal bioactive entities identified are MAMPs, such as lipoteichoic acid (LTA), peptidoglycan (PGN), exopolysaccharide (EPS), and surface proteins.

These molecules were recognized by PRRs expressed on epithelial immune cells, including Toll-like receptors (TLRs) and NOD-like receptors (e.g., NOD2). Signal transduction predominantly occurred via the MyD88-dependent pathway [15,49,50], activating NF-κB and MAPKs such as ERK, JNK, and p38, concurrently contributing in the regulation of cytokine production [51,52,53], in DC activation, and in the balance of T_h_1, T_h_2, T_h_17, and T_reg_ lymphocytes [15].

In particular, a treatment with micronized and heat-treated *L. plantarum* was shown to upregulate NO and cytokine production through the TLR2/MyD88/MAPKs/NF-κB axis [30]. Similarly, a strain of *L. sakei* activated NF-κB via phosphorylation, and the degradation of its inhibitor IκBα promoted Extracellular Signal-Regulated Kinases (ERKs), c-Jun N-terminal kinase-dependent (JNK), and MAPKp38 phosphorylation in macrophages [41].

Ultimately, lactobacilli-derived MAMPs play a dominant role in modulating the immune response, directing both pro-inflammatory and anti-inflammatory pathways depending on the host status and receptor engagement. These findings offer a mechanistic rationale for the use of non-viable lactobacilli in immune-related therapeutic applications.

### 3.3. Host Determinants of the Pro-/Anti-Inflammatory “Switch”

A single MAMP can yield opposite outcomes depending on the host context. Below, the host-side determinants gate this switch and thereby reconcile pro-inflammatory effects in resting tissues with pro-resolving effects under inflamed conditions.

#### 3.3.1. PRR Abundance and Subcellular Routing

Inflammation reshapes the PRR landscape—e.g., the epithelial and myeloid upregulation of TLR2 and altered NOD2 transcription—amplifying MAMP sensitivity but also changing which pathways are available. In parallel, receptor trafficking dictates signal bias: TLR4 initiates MyD88 signaling at the plasma membrane and TRIF signaling after endocytosis [54,55]. TLR2 can also signal from endosomes to IRF7/IFN-I when sorted via TRAM [56]. In the gut epithelium, low TLR4/MD-2 and basolateral restriction further shape outputs [57]. Together, expression and routing re-weight cytokine programs.

#### 3.3.2. Pre-Activation State: Tolerance vs. Trained Immunity

Prior stimulation imprints innate cells via chromatin and metabolic rewiring. Endotoxin tolerance represses selected NF-κB target loci and limits cytokines [58], whereas trained immunity installs activating marks and glycolytic programs that potentiate responses to subsequent triggers [59]. The prevailing imprint (tolerance or training) biases whether a given MAMP dampens or boosts inflammation.

#### 3.3.3. Adaptors and PRR Crosstalk; Inflammasome Priming

Signal integration at adaptor hubs (MyD88 vs. TRIF) and PRR cooperation (e.g., Dectin-1-TLR2 [60], NOD2-TLR2 [61]) qualitatively remodel transcriptional programs beyond simple additivity. In addition, NLRP3 requires “signal-1” priming (often via TLR/NOD) before “signal-2” activation; the presence or absence of priming strongly conditions IL-1β/IL-18 outputs for the same MAMP [62].

#### 3.3.4. Endogenous Negative Regulators

Cells in inflamed milieus frequently upregulate intracellular brakes, IRAK-M/IRAK3 [63], A20/TNFAIP3 [64], SIGIRR/IL-1R8 [65], and TOLLIP [66]. These molecules limit MyD88 complex assembly, edit ubiquitin chains in NF-κB cascades, or dampen IL-1R/TLR signaling, thereby curbing cytokine amplitude even when PRRs are abundant.

#### 3.3.5. Immunometabolic Set-Point

Resting/repairing programs (OXPHOS, fatty acid oxidation) favor regulatory outputs, whereas glycolysis supports rapid pro-inflammatory effector functions [67]. Metabolites such as lactate and ROS further re-tune MAPK/NF-κB—lactate (via GPR81/YAP) suppresses NF-κB-dependent cytokines [68,69]; ROS can either activate or restrain NF-κB depending on the concentration and compartment [70]. Thus, the metabolic state steers the same MAMP toward resolution or escalation.

#### 3.3.6. Cytokine Milieu

Pre-existing IL-10 and TGF-β skew responses toward regulatory phenotypes (e.g., promotion/maintenance of Foxp3^+^ T_reg_), reducing epithelial chemokines and restraining myeloid TNF/IL-12 despite ongoing PRR engagement [71]. Tissue-derived retinoic acid cooperates with TGF-β in the gut to stabilize this regulatory bias [72].

#### 3.3.7. Cell Type-Specific Signal Decoding

The same MAMP is interpreted differently by intestinal epithelial cells (IEC) versus dendritic cells/macrophages due to distinct PRR repertoires, adaptor usage, and chromatin landscapes. IECs often prioritize barrier-protective programs (e.g., TLR2-dependent tight junction support) and limit TLR4 sensing [73], while DC subsets stratify PRRs to shape T-cell outcomes [74]; stimulus-induced “latent enhancers” in macrophages further encode history-dependent responses [75].

## 4. Lactobacilli Acts on PRR Pathways to Exert Immuno-Regulating Effects

Many studies have reported the ability of lactobacilli to determine immunomodulatory effects, directly or indirectly, by regulating PRR-dependent signaling pathways, sometimes acting even at a transcriptional level. They have been found to regulate, in particular, TLR (for instance, TLR-1/2 [30,76]; TLR4 [77]; TLR7 [78]; TLR-9 [78,79]) and NLR pathways (for instance, NLRP3 [80,81,82]; NLRC4 [82]; NOD1 [83]). However, the complexity of PRR-MAMP recognition and the signaling effectors involved are still yet to be discovered in their full extent. *Ligilactobacillus salivarius* UCC118 can stimulate TLR2 and TLR4-independent, but MyD88-dependent stimulation of cytokines (TNF-α, IL-6, IL-10, IL-12p70, IL-1β) in bone marrow-derived macrophages (BMDMs); in this case, the TLR-independent mechanism was associated with another receptor, Clec4e (Mincle), a CLR which is supposed to have functional associations with TLR2 in both humans and mice [84]. This demonstrates the great variance of PRRs whose function is regulated by probiotics, lactobacilli, and their components in particular.

### 4.1. General Overview of PRRs

The immune recognition of MAMPs was mediated by pattern recognition receptors (PRRs), expressed on the surface or within intracellular compartments of both immune (e.g., macrophages, DCs, NK cells, monocytes, neutrophils, T and B lymphocytes) and non-immune cells (e.g., epithelial cells, endothelial cells, fibroblast, smooth muscle cells) [85,86,87]. Some PRRs were even found in soluble form, circulating in biological fluids [88]. These receptors detected conserved MAMPs, which were critical from microbial surveillance and highly conserved within specific microbial groups. Five main PRR families were recognized based on structure, location, and ligand specificity: Toll-like receptors (TLRs), C-type lectin receptors (CLRs), NOD-like receptors (NLRs), RIG-I-like receptors (RLRs), and AIM2-like receptors (ALRs) [85,86,89,90]. Among these, TLRs and NLRs were the most thoroughly characterized in host–microbe interactions and are therefore discussed in greater detail below.

### 4.2. Toll-like Receptors (TLRs)

#### 4.2.1. Structure and Families

TLRs represented type I transmembrane glycoproteins composed of an extracellular leucine-rich repeat (LRR) domain for ligand recognition, a single transmembrane helix, and a cytoplasmic Toll/IL-1 receptor (TIR) domain responsible for downstream signal transduction [85,87]. The LRR motifs, repeated 19–25 times, formed a horseshoe-shaped structure that facilitated selective MAMP binding through conserved residues on the concave surface and variable residues on the convex surface [86]. TLRs recognized not only MAMPs but also damage-associated molecular patterns (DAMPs) and xenobiotic-derived molecular patterns (XAMPs) [89].

Thirteen TLRs were identified in mammals, of which ten (TLR1-10) were expressed in humans, and TLR11-13 only in mice [91,92,93]. While TLR1, TLR2, TLR4, TLR5, TLR6, and TLR10 were membrane-bound and sensed extracellular microbial structures (e.g., lipoproteins, LPS, flagellin), intracellular TLRs (TLR3, TLR7, TLR8, TLR9, TLR11-13) were localized in endosomal and lysosomal compartments and specialized in nucleic acid sensing [94].

#### 4.2.2. Cellular Distribution and Ligands

TLRs were broadly expressed in antigen-presenting cells (APC), granulocytes, lymphocytes, epithelial and endothelial cells, and tissue-resident stromal cells [88,93]. As shown in Figure 2, TLR2 formed heterodimers with TLR1 or TLR6 to expand its ligand repertoire, which included LTAs, LPPs, and PGNs [61,95,96]. TLR3 recognized dsRNA [97] and TLR4 recognized LPS from Gram-negative bacteria [93], whereas TLR5 detected flagellin [98]. TLR7 and TLR8 recognize GU-rich and AU-rich sequences on ssRNA, typically of viral origin [99], and TLR9 senses unmethylated CpG motifs in bacterial DNA [100]. Complex receptor trafficking, including endocytosis and recycling, contributed to the spatial and functional regulation of TLR activation [87,89]. Some of the TLRs were mainly expressed on the cell surface (TLR-1, -2, -4, -5, -6, -10), where they recognize foreign structural components, while others (TLR -3, -7, -8, -9- 11, -12, -13) were typically found in the endoplasmic reticulum, in endosomes, and in lysosomes [91,93].

#### 4.2.3. Signal Transduction

Upon ligand binding and dimerization, TLRs undergo conformational changes in the TIR domain, facilitating adaptor recruitment. Most TLRs used the MyD88-dependent pathway [87], except TLR3, which exclusively employed the TRIF-dependent pathway [89]. MyD88 contained a death domain (DD) that recruited IL-1 receptor-associated kinases (IRAK-1, -2, -4), leading to myddosome assembly and the downstream activation of TRAF6, TAK1, and the MAPK/NF-κB pathways [92,101]. The IKK complex (IKKα, IKKβ, and NEMO) enabled IκBα degradation and the subsequent nuclear translocation of NF-κB [102].

TRIF-dependent signaling, used by TLR3 and TLR4, activated IRF3 and type I interferon expression through TRAF3 and TBK1/IKK-ε, which regulated the expression of IFN-α and IFN-β to induce an antiviral immune response [97,103]. Some TLRs, such as TLR7-9, also trigger IRF7 activation in plasmacytoid DCs, promoting antiviral IFN-α production [104]. Negative regulators (e.g., TOLLIP, IRAK-M, and SOCS1) controlled TLR signaling by dampening cytokine production [96].

TLR engagement shapes the adaptive immune response by promoting DC maturation, antigen presentation (MHC class II, co-stimulatory molecules), and cytokine-driven T_h_ polarization [88]. TLRs thus act as critical molecular links between microbial detection and immunological memory.

### 4.3. NOD-like Receptors (NLRs)

#### 4.3.1. Structure and Classification

NLRs represent cytosolic PRRs that monitor intracellular compartments for microbial molecules and stress signals. They share a tripartite structure composed of a C-terminal LRR domain for ligand recognition, a central NACHT (NOD) domain for ATP-dependent oligomerization, and a variable N-terminal domain that defines the subfamilies: an acidic transactivation domain (AD domain) or NLRA, an NLRB or baculoviral inhibitory repeat-like domain (BIR domain), an NLRC or caspase activation and recruitment domain (CARD domain), and an NLRP or pyrin domain (PYD domain) [102]. A total of 23 NLRs were identified in humans, expressed by both immune and non-immune cells [105,106,107].

#### 4.3.2. Ligand Recognition and Inflammasome Formation

Certain NLPRs, such as NLRP1, NLRP3, NLRC4, and NAIP, formed multiprotein complexes called inflammasomes upon activation. These complexes consisted of a sensor NLR, the ASC adaptor (PYD-CARD), and pro-caspase-1, which, upon activation, cleaved pro-IL-1β and pro-IL-18 into mature cytokines [108]. Ligands included muramyl dipeptide (MDP, NOD2), iE-DAP (NOD1), flagellin (NLRC4/NAIP), LTA (NLRP6), and bacterial lipopeptides (NLRP7) [109,110]. Beyond canonical inflammasomes, some NLRs functioned as signaling hubs or negative regulators. NLRP10 and NLRP12 modulated DC activity and adaptive immunity initiation [111], while NLRP2/4 and NLRC3 downregulated NF-κB via TRAF6 inhibition [110].

#### 4.3.3. Signal Transduction and Effector Activation

NOD1 and NOD2, upon sensing iE-DAP or MDP, respectively, recruited RIPK2 via CARD-CARD interactions—RIPK2 underwent K63-linked ubiquitination and triggered the TAK1-dependent activation of NF-κB, MAPK (p38, JNK, ERK), and cytokine/chemokine expression [112,113]. These pathways facilitated early immune defense and leukocyte recruitment. Unlike TLRs, NLR activation was independent of membrane trafficking and relied solely on cytoplasmic sensing mechanisms.

TLRs and NLRs represented the principal PRRs through which the host immune system detected microbial signals. While TLRs were specialized in sensing extracellular and endosomal ligands via transmembrane signaling, NLRs monitored cytosolic perturbations and enabled rapid responses to invasive microbes. Both pathways converged on similar effectors (NF-κB, MAPKs, IRFs), yet their compartmentalized activation and ligand specificity conferred complementary roles in immune surveillance. Understanding the differential engagement of TLRs and NLRs by lactobacilli-derived MAMPs offered a deeper comprehension into probiotic immunomodulation.

## 5. MAMPs of Lactobacilli and Their Immunomodulatory Effects

Lactobacilli MAMPs can interact with PRRs, mediating multiple immunological responses. A list of key molecular patterns from different lactobacilli species and their immunomodulatory effects is presented below and also described in more detail in Appendix A.

### 5.1. Peptidoglycan (PGN)

#### 5.1.1. Signal Transduction and Effector Activation

PGN represents a key structural element of the lactobacilli cell wall, contributing to mechanical strength, cellular morphology, and adhesion properties [114]. Alongside polyanionic teichoic acids (TAs) and surface proteins, PGN conferred elasticity, porosity, tensile strength, and electrostatic potential to the bacterial envelope, supporting ion homeostasis and the trafficking of nutrients, proteins, and antibiotics [115]. As shown in Figure 3, its structure included repeating disaccharide units of N-acetylglucosamine (GlcNAc) and N-acetylmuramic acid (MurNAc), cross-linked by short peptide chains containing both D- and L-aminoacids [114,116,117]. PGN is recognized by host PRRs [118,119], such as TLR2 and NOD2 [120,121], triggering immune signaling cascades such as NF-κB and MAPK [122]. In vivo murine model studies demonstrated that PGN derived from lactobacilli upregulated TLR2 and MAPKp38 expression in macrophages and modulated gene expression by enhancing stress response and antigen presentation markers including IL-12 and MHC II, potentially facilitating T_h_1-type immune polarization [123].

#### 5.1.2. Species- and Strain-Specific Effects

Among the best characterized strains, *L. rhamnosus* CRL1505-derived PGN suppressed HLA-DR, CD80, and CD83 expression in LPS-induced DCs while upregulating the increase simultaneously of TNF-α and IL-10 in LPS-stimulated DCs, exerting an anti-inflammatory activity. Furthermore, it increased CD86 in DCs under basal conditions [124]. It also downregulated IL-1β, IL-6, and TNF-α in LPS-stimulated RAW 264.7 macrophages [119]. Furthermore, in avian splenocytes, high doses of PGN from *L. rhamnosus* MLGA reduced IL-8 and IL-1β, even in the absence of prior stimulation; lysozyme-digested PGN fragments also upregulated β-defensin-9 in chicken peripheral blood mononuclear cells (PBMCs) and other tissues [125]. In murine models of *Streptococcus pneumoniae* infection, PGN treatment enhanced leucocyte recruitment, increased mucosal and systemic cytokines (TNF-α, IL-10, IL-6, and IL-1β), and upregulated immunoglobulin (IgA, IgG, and IgM) production. It also upregulated TLR2 and TLR-9 expression in alveolar and peritoneal macrophages, thus increasing pathogen recognition capacity [126,127]. In dual infection models with respiratory syncytial virus (RSV) and pneumococcus, PGN from CRL1505 reduced both viral load and bacterial burden [128]. In poly (I:C)-primed mice, it downregulated TLR-3 activation, suppressed IL-6, and upregulated IFN-α, -β, -γ, TNF-α, and IL-10 [129,130], suggesting a role in antiviral and anti-inflammatory defense. In this context, alveolar macrophages (AMs) played a basic role: when isolated from treated mice, they produced in vitro higher levels of IFN-γ, TNF-α, IL-6, IL-27, and chemokines such as CCL2, CXCL2, and CXCL10, enhancing immune cell recruitment [128]. Similarly, *L. paracasei*-derived PGN was also shown to significantly reduce the viral load of influenza virus in the lungs of infected mice. This antiviral effect was associated with the increased recruitment of DCs to the respiratory mucosa, paralleled by a reduction in the proportion of other immune cell subsets, suggesting a reshaping of the local immune milieu to favor antigen presentation and effective viral clearance [131].

PGN from other strains also showed immunomodulatory activity. Indeed, PGN derived from *Lactobacillus acidophilus* KLDS 1.0738 promoted regulatory T-cell responses, reducing IgE and enhancing IL-10, IFN-γ, and TGF-β secretion in β-lactoglobulin-sensitized mice [132]. In LPS-stimulated RAW264.7 cell lines, it reduced iNOS, COX2, and pro-inflammatory cytokines (IL-6, IL-1β, TN-α), highlighting anti-inflammatory properties [119,133]. Interestingly, similar effects were observed for some strains of *L. plantarum*: the PGN derived from ATCC 14917 suppressed IL-8 in poly (I:C)-stimulated HT-29 cells in vitro [47], and PGN from *L. plantarum* CAU1055 reduced levels of iNOS, COX2, and pro-inflammatory cytokines (IL-6, TNF-α) in mice models [134]. Conversely, PGN from *L. plantarum* AR113 increased the production of pro-inflammatory cytokines in RAW264.7 cells [135].

Several studies also examined PGN-derived fragments like muramyl dipeptide (MDP), a NOD2 ligand, which exerts an immunosuppressive effect by attenuating TLR4-mediated signaling [136]. In LPS-stimulated RAW 264.7 cells, MDP from *L. reuteri* inhibited IL-1β, IL-6, and CCL20 by suppressing PI3K/Akt, MAPK, and TLR4 expression [137]. A GlcNAc-MurNAc tripeptide (GMTP) from *L. acidophilus* strongly induced COX2, IL-1β, and TNF-α expression, highlighting the key immunostimulatory role of GlcNAc residues [118].

Moreover, polysaccharide–peptidoglycan complexes (PSPGs) from *L. casei* Shirota inhibited IL-6 production in lamina propria mononuclear cells via NOD2 activation and NF-κB suppression [138]; these effects were also confirmed in vivo [139]. Peptidoglycan hydrolases (PGHs), such as p40 and p75, modulated immune response by releasing PGN fragments. In DCs, Lc-p75-defective *L. casei* exhibited an impaired uptake and reduced production of TNF-α, IL-1β, and IL-6 compared to the WT [140]. A bifunctional PGH named LPH, derived from lactobacilli, attenuated TNBS-induced colitis in mice by digesting PGN and releasing NOD2 ligands, thereby increasing IL-10 and decreasing TNF-α, IFN-γ, and IL-6, exerting an anti-inflammatory activity [24].

Overall, the immunomodulatory activity of lactobacilli-derived PGN appears to be profoundly influenced by dose, structural specificity, and temporal dynamics. Dose-dependent effects have been widely reported, impacting both the amplitude and the cytokine profile of the immune response. Likewise, structural variations, such as the presence of immunogenic muropeptides like M-tri-Lys or differential D-Ala termination, confer species- and strain-specific signatures that modulate immune recognition and downstream signaling. Moreover, the kinetics of PGN-mediated activation are not uniform: early responses typically involve IL-1, IL-6, IL-12, and co-stimulatory molecules (CD80, CD86), while TNF-α expression often requires sustained stimulation [123,126,130,134,139,141]. These findings underscore the need to consider quantitative, structural, and temporal parameters when evaluating PGN-based immunomodulatory interventions.

### 5.2. Lipoteichoic Acids (LTAs)

#### 5.2.1. Structure and Key Features

LTA, a key component of Gram-positive cell walls, consists of amphiphilic molecules anchored to the cytoplasmic membrane via glycolipids and extended outward with polyglycerophosphate chains rich in D-Ala substitutions [115] (Figure 4). In lactobacilli, LTA functions as one of the major MAMPs, recognized primarily by TLR2/TLR6 heterodimers but also involving scavenger receptors such as SR-A [142]. Studies have demonstrated that LTA recognition triggered different NF-κB pathways [143,144], MAPKs [145,146,147,148], PI3K-Akt, cPLA2, and STAT1/JAK1 in macrophages and intestinal epithelial cells [147,148,149,150,151]. Some host receptors different than TLR2, which are still to be identified, may be involved in some of the LTA effects [152]. Depending on the inflammatory context, LTA exhibited dual immunomodulatory functions. The in vitro pre-treatment of macrophages or intestinal epithelium cells with LTA from various lactobacilli strains attenuated the phosphorylation of pro-inflammatory transducers (STAT-1, JAK-1, NF-κB, ERK, JNK, p38, HSP27), reduced IκBα degradation, and suppressed IL-6, IL-8, and TNF-α [148,149,150,151,153,154]. The immunological effects of LTA were strongly influenced by its structure. The acylation of LTA was necessary to induce NF-κB activation and IL-8 expression in Caco-2 cells and RAW264.7 cells, whereas deactivated LTA lost its immunostimulatory potential [144,155]. D-alanylation also played a crucial regulatory role: mutants lacking D-Ala residues in *L. plantarum* CRL1506 and WCFS1 and *L. casei* BL580 showed a lower immunomodulatory activity and altered TLR expression profiles in models of DSS (dextran sodium sulfate)-induced colitis and poly (I:C) challenge [152,156,157]. In LGG, the deprivation of D-Ala residues on LTA yielded a distinct phenotype: in the DSS-induced colitis mouse model, the modified LTA elicited stronger immunomodulatory effects than the wild-type LTA [144,158].

#### 5.2.2. Species- and Strain-Specific Effects

Several studies confirmed that LTA-mediated effects varied between species: *L. plantarum* LTA, in both diverse in vitro and in vivo models, reduced IL-8, TNF-α, IL-12, IL-4, IL-1β, and IL-6 [150,151,154,159,160,161], while enhancing IL-10 production [162]. These effects are not common in all probiotics; for example, IL-8 inhibition is typical of *L. plantarum*-derived LTA but not of *L. rhamnosus*, *Lactobacillus delbrueckii*, and *L. sakei* LTA, which showed minor effects in the same models [163]. Differences in acyl chains and D-Ala content were implicated as drivers of this specificity [142,153]. Furthermore, LTA from *L. plantarum* and from *L. casei* enhanced IL-10 while suppressing IL-12 in macrophages in vitro [145], but *L. plantarum* strain K8-LTA instead stimulated TNF-α production in PGE/LPS-primed THP-1 cells, highlighting the bidirectionality of LTA signaling [164].

*L. plantarum*-LTA reduced the expression of adhesion molecules (ICAM-1, VCAM-1, and E-selectin) in HUVEC and epithelial cells, blocked LPS-induced NO and COX2 in macrophages, and modulated cytokine responses during ETEC infection in piglets [150,159]. Furthermore, LTA from LGG restored antigen presentation and T-cell priming capacity in UVB-immunosuppressed mice, suggesting a broad immunostimulatory potential [165]. In the intestinal setting, LTA from *Lacticaseibacillus rhamnosus* GG also reprogrammed gut dendritic cells and T-cell responses, further supporting its role as a mucosal immunomodulator [166]

LTA-deficient strains such as *L. acidophilus* NCK2025 offered crucial comprehension into the immunobiology of LTA. These mutants reduced the surface expression of MHC-II, CD40, CD80, and CD86 and decreased the production of TNF-α, IL-6, and IL-12, while increasing IL-10 in a time-dependent manner [147,167,168]. In colitis models, NCK2025 treatment improved disease markers and elevated T_reg_ population, underscoring the pro-inflammatory role of WT-LTA. Similarly, *L. paracasei* LTA reduced TNF-α, IFN-γ, IL-6, and IL-1β expression, thereby improving colonic inflammation [146,169].

### 5.3. Exopolysaccharides (EPSs)

#### 5.3.1. Structure and Key Features

EPSs produced by lactobacilli are high-molecular-weight (10 to 1000 kDa) carbohydrate polymers secreted into the extracellular matrix. Depending on their association with the bacterial surface, they are classified as capsular EPSs, when tightly bound, or as slime EPSs, when loosely associated. Structurally, EPSs vary significantly in molecular weight, charge, degree of branching, and types of glycosidic linkages. This molecular heterogenicity influences both their physiochemical properties and their biological activity [170,171,172].

The immunological and antioxidant effects of EPSs depend not only on their molecular weight, but also on more refined features such as monosaccharide composition, structural conformation, and substitution patterns. While low-molecular-weight EPSs were generally more bioactive [173,174], several reports indicate that higher-molecular-weight EPSs showed enhanced immunoregulatory functions [175,176]. Acidic EPSs, in particular, were found to possess a greater antioxidant and immunostimulatory potential than neutral EPSs, likely due to an increased electron-donating ability and metal ion chelation [177,178].

The immunomodulatory effects of EPSs were mediated by several host signaling pathways: several studies demonstrated the involvement of NF-κB, MAPKs (ERK, JNK, p38), STAT3, MyD88, and c-Jun in EPS-induced immune responses [179,180,181]. In models using LGG-derived EPS, the phosphorylation of IκB was inhibited and the nuclear translocation of p65 was reduced, leading to decreased inflammatory signaling [182]. TLR2, TLR4, and TLR-5 were among the primary receptors affected, while MyD88 played a central role in downstream signal transduction [183,184]; these data are schematically summarized in Figure 5. In hepatic inflammation models, the IL-17/TLR2/p38/STAT axis was particularly implicated in *L. plantarum* EPS-mediated protective effects [180].

EPSs exhibited robust antioxidant properties by scavenging hydroxyl, superoxide, and DPPH radicals in a concentration-dependent manner [185,186]. In vitro assays on macrophage, enterocyte, and neuronal cells revealed that EPSs enhanced the total antioxidant capacity (T-AOC) and increased the activities of key enzymes such as superoxide dismutase (SOD), catalase (CAT), and glutathione peroxidase (GSH-Px). These effects contributed to a protection against H_2_O_2_-induced oxidative stress [187,188,189]. Furthermore, an EPS pre-treatment of epithelial cell cultures preserved tight junction proteins (claudin-1, occludin, ZO-1), counteracting the deleterious effects of TNF-α and IL-1β on epithelial barrier integrity [175].

EPSs modulated macrophage function by stimulating proliferation, enhancing phagocytosis, and increasing the production of NO, ROS, and cytokines (TNF-α, IL-1β, and IL-6, IL-10), often in a dose-dependent manner [190,191,192,193]. In HT-29 epithelial cells, EPSs induced TNF-α and IL-1β, while promoting moderate levels of IL-4 and IL-10 [194]. In models of immune stimulation (e.g., LPS-treated macrophages), EPSs showed anti-inflammatory properties by downregulating ROS production, inhibiting iNOS and COX2 activity, and reducing the secretion of pro-inflammatory cytokines [179,187]. In dermal fibroblast, EPSs reduced inflammatory mediators, suggesting potential applications in cutaneous inflammation [175].

EPSs alleviated gut inflammation in multiple murine models, including DSS-, AOM/DSS-, and pathogen-induced colitis. The treatment resulted in reduced levels of TNF-α, IL-1β, IL-6, IL-8, IFN-γ, and IL-17, while upregulating anti-inflammatory cytokines IL-10 and IL-4. Moreover, EPSs restored tight junction protein expression, ameliorated crypt damage, and also prevented colon tumor development and improved disease activity index scores [195,196,197,198,199].

#### 5.3.2. Species- and Strain-Specific Effects

In RAW264.7 macrophages, *L. rhamnosus*-derived EPSs significantly increased phagocytic activity and promoted the secretion of NO, TNF-α, IL-1β, IL-6, and IL-10 in a dose-dependent manner [200,201], whereas EPS from the KL37C strain failed to stimulate cytokine production or phagocytosis but reduced ROS levels, suggesting antioxidant effects [202].

EPSs from *L. acidophilus* induced IL-1α, CCL2, TNF-α, and PTX3 in Caco-2 cells, indicating an enhanced mucosal immune priming [203]. EPSs from *L. plantarum* promoted dendritic cell maturation by increasing MHC-II and CD86 expression, elevating IL-12p70, and suppressing IL-10, thereby skewing responses toward a Th1 phenotype [204]. By contrast, EPSs from *L. casei* did not affect NO or IL-6 in unstimulated RAW264.7 macrophages, but increased TNF-α in a dose-dependent manner; in the same model, LPS-induced EPS exposure decreased NO production, indicating a context-dependent immunomodulation [181].

The strain-dependent variability was also evident in animal models: in colitis-induced mice models, *L. plantarum*-derived EPSs exhibited strong protective effects against chemically induced colitis in mice, outperforming EPSs from *L. rhamnosus* [199]. Conversely, *L. plantarum* T10 failed to reduce IL-6 levels in colitis models, while other *L. plantarum* strains showed potent anti-inflammatory effects [198]. Nonetheless, *L. rhamnosus*-derived EPSs improved disease parameters such as body weight loss, epithelial integrity (ZO-1, occluding, claudin-1), and crypt morphology [205,206]. In infectious models, the same EPSs reduced *Salmonella typhimurium*-induced intestinal damage and decreased pro-inflammatory cytokines and IL-4 levels [182], while also exerting neuroprotective effects against oxidative injury [205] and anti-allergic properties by suppressing T_h_2 cytokines (IL-4, IL5, IL-13) and eosinophilic infiltration in sensitized mice [207]. Notably, *L. acidophilus* EPSs reduced hepatic inflammation by modulating IL-17, TGF-β, IL-10, and glutathione levels and downregulating ALT and γ-GT serum markers [208]. Moreover, EPS from *L. delbrueckii* spp. *bulgaricus* similarly induced IL-6, IL-10, IFN-γ, and NK activity both in vitro and in vivo [178]. Furthermore, *L. paracasei* mitigated colitis through MIP-2 suppression and IL-10 upregulation [209].

EPSs from *L. reuteri* L26 and DSM17938 enhanced DC activation and MHC-II/CDC80 expression and modulated IL-1β, IL-6, IL-10, IL-12p35, and TGF-β both in vivo and in vitro [184,210]. Anti-inflammatory and anti-adhesion properties were also confirmed in ETEC-infected enterocytes [211].

Notably, EPSs produced by lactobacilli also exhibit significant antioxidant properties, thereby complementing their immunomodulatory profile; for instance, EPSs derived from *L. plantarum* demonstrated a dose-dependent antioxidant activity, including an enhanced radical scavenging and upregulation of endogenous antioxidant enzymes, such as SOD and CAT [185,212]. A similar antioxidant potential was exerted also by EPSs from *L. rhamnosus* [182,205], *L. casei* [181], *L. paracasei* [213], and *L. delbrueckii* spp. *bulgaricus* [174].

### 5.4. S-Layer Proteins (SLPs)

#### 5.4.1. Structure and Key Features

SLPs constitute the outermost proteinaceous layer of the bacteria cell envelope in many lactobacilli species. These proteins, which varied in molecular weight, structure, and post-translation modifications, formed a para-crystalline monolayer bound covalently to the cell wall polysaccharide (Figure 6) [214]. SLPs contributed to protection against environmental stressors, including acidic gastric conditions, as demonstrated by the reduced survival of *L. acidophilus* upon SLP removal [215]. Additionally, SLPs promoted bacterial adhesion and auto- and co-aggregation, supporting the microbial colonization of mucosal surfaces [216].

SLPs modulated host immunity by interacting with immune cells and influencing cytokine production in a species- and strain-dependent manner [217]. In LPS-stimulated RAW264.7 macrophages, SLPs purified from *L. acidophilus* NCFM downregulated iNOS and COX2 expression, resulting in reduced NO and prostaglandin E2 levels, while also decreasing TNF-α and IL-1β secretion [218]. In Caco-2 cells, NCFM SLPs attenuated TNF-α-induced IL-8 production and prevented the disruption of tight junction proteins ZO-1 and occludin [219].

SLP-deficient mutants of NCFM (e.g., ΔslpB, ΔslpX and ΔslpB) displayed altered immunostimulatory profiles. A co-culture with bone marrow DCs revealed a reduced expression of MHC-II, CD11b, and cytokines such as IL-6, MIP-1α, and IL12p40 compared to WT strains [220]. These findings were consistent with earlier studies using engineered strains such as NCK1377-Cl (SlpA-knockout/SlpB-dominant), which elicited higher IL-12p70, TNF-α, and IL-1β but reduced IL-10 levels compared to WT NCFM. Otherwise, IL-6 was strongly prompted by the WT strain compared to the mutant [221]. Other studies further confirmed that mutants lacking SLPs induced weaker T_h_1 cytokine responses and promoted IL-10 production in DCs [222,223].

#### 5.4.2. Species- and Specific-Strain Effects

Several SLPs engaged PRRs to initiate immune signaling. SlpA from *L. acidophilus* bound to DC-SIGN on DCs, regulating both pro- and anti-inflammatory cytokine production. In the presence of LPS, SlpA promoted a T_h_2-skewed response through IL-4 production, although SlpA alone was insufficient to stimulate naïve CD4^+^ T-cell proliferation [221,224]. SLPs also modulated MAPK pathways: SLPs from *L. acidophilus* ATCC4356 reduced ERK1/2, JNK, and p38 phosphorylation in Caco-2 cells challenged with *Salmonella typhimurium*, while simultaneously downregulating IL-8 secretion and protecting tight junction integrity [225,226].

SLPs from *L. acidophilus* ATCC4356 induced CD80 and CD86 expression in DCs and promoted IL-10 secretion, while decreasing IFN-γ and TNF-α production. They also enhanced interferon-stimulated gene (ISG) expression, suggesting a role in antiviral defense [227]. In macrophages, *L. acidophilus* CICC6074 SLPs increased phagocytic activity and NO production in a dose-dependent manner [215]. Similar immunostimulatory effects were observed for SLPs from *L. helveticus* LH2171, which upregulated hBD expression in Caco-2 cells via TLR2 and JNK activation. Other species, including *Limosilactobacillus amylovorus*, *Lentilactobacillus buchneri*, and *Levilactobacillus brevis* demonstrated a comparable but strain-specific JNK pathway activation [228].

SLPs from LGG decreased TNF-α and IL-6 production in RAW264.7 macrophages and inhibited ERK phosphorylation following LPS challenge [229]. In IPEC-J2 epithelial cells, LGG-derived SLPs and EPSs reduced the transcription of pro-inflammatory cytokines (IL-6, IL-12, and TNF-α) [230]. SLPs from *L. paracasei* spp. *paracasei* M5-L and *L. casei* Q8 attenuated pathogen-induced reductions in ZO-1 and occludin in the gut epithelium, protecting epithelial integrity [231,232]. Similarly, *L. plantarum* SLPs alleviated IL-8 and TNF-α responses in inflamed Caco-2 cells and restored tight junction expression by modulating LPS-induced miRNA dysregulation [233].

A domain of an SLP, termed the micro integral membrane protein (MIMP), isolated from *L. plantarum* CGMCC1258, significantly reduced inflammation in DSS-induced colitis mice by decreasing IFN-γ, IL-17, and IL-23 while increasing IL-4 and IL-10, and was also found to improve the health status of these mice, while at the same time reducing IFN-γ, IL-17, and IL-23 levels and up-expressing IL-4 and IL-10 levels compared to the mice who did not receive MIMP treatment [234]. These anti-inflammatory effects were also confirmed by a PBMCs/Caco-2 cell co-culture model in vitro challenged with LPS, where it has been established that MIMPs interfere with the TLR4 pathway and with NF-κB, MAPK, and JNK. Also, important effects on histone acetylation were discovered, and it was reported that MIMPs can inhibit LPS-induced histone acetylation, modulating cell transcription at an epigenetic level [235].

Despite the strong evidence for strain-specific activity and pathway modulation, the complete spectrum of SLP–host receptor interactions remained incompletely defined. Further research is needed to elucidate the structure–function relationship of SLPs and to explore their therapeutic applications in inflammation and mucosal immunity.

### 5.5. Pili

#### 5.5.1. Structure and Key Features

Pili or fimbriae represent long filamentous structures protruding from the bacterial cell wall, composed of polymerized pilin subunits arranged via a sortase-dependent mechanism [236]. These structures facilitated adhesion to host tissues, interbacterial aggregation, and environmental sensing. Among lactobacilli, the SpaCBA pilus system of LGG was the most characterized [237,238] (Figure 7).

#### 5.5.2. Species- and Strain-Specific Effect

The SpaCBA pilus, and particularly the SpaC adhesin loaded at its tip, mediated high-affinity binding to intestinal mucins via β-galactoside-containing carbohydrate motifs [239,240,241]. This adhesion promoted the colonization of intestinal epithelial cells and mucus layers and was essential for SpaC-dependent immunological signaling. The deletion of SpaC impaired mucin binding and reduced LGG’s ability to colonize the intestinal surface [240].

SpaCBA pili also functioned as immunomodulatory structures: in human monocyte-derived DCs, purified SpaCBA pili activated TLR2-dependent NF-κB signaling and triggered cytokine production, including IL-6, IL-10, and IL-12 [240,242]. Interaction with the DC-SIGN receptor on immature DCs further supported SpaCBA-mediated immune engagement, inducing IL-6, IL-10, IL-12p40, and IL-12p35. In SpaCBA-deficient LGG mutants, Caco-2 cells exhibited an increased IL-8 expression, and macrophages showed decreased IL-10 and elevated IL-6 production, confirming the role of SpaCBA in balancing pro- and anti-inflammatory responses [239,243].

SpaCBA-expressing LGG enhanced ERK phosphorylation and ROS generation in epithelial cells, promoting cell proliferation and exhibiting radioprotective effects both in vitro and in vivo; these functions were absent in SpaC-deficient mutants [244]. Moreover, SpaC contributed to the competitive exclusion of *Enterococcus faecium* by preventing its adhesion to intestinal mucus, indicating a potential role in microbiota modulation and pathogen control [245].

The SpaCBA pilus system of LGG exemplified a multifunctional bacterial appendage that combined adhesive and immunological functions. Through the direct engagement of epithelial and immune cells, SpaCBA pili modulated host responses, reinforced mucosal integrity, and potentially contributed to host defense against pathogens. Future studies should explore additional pili/fimbriae in lactobacilli to determine whether similar dual-function MAMP systems are widespread across probiotic species.

### 5.6. Lipoproteins (LPPs)

#### 5.6.1. Structure and Key Features

Bacterial LPPs are post-translationally modified proteins covalently anchored to the cytoplasmic membrane through lipid moieties. In Gram-positive bacteria, including lactobacilli, these lipoproteins are surface-exposed and accessible to host immune sensors [246,247]. Their immunological relevance stems from their capacity to act as MAMPs, particularly through the engagement of TLR2. Indeed, TLR2 forms heterodimers with either TLR1 or TLR6, allowing for the recognition of tri- and di-acetylated LPPs, respectively [248,249]. Despite this well-established signaling paradigm, LPPs from probiotic lactobacilli remain relatively underexplored in terms of molecular specificity, structure–activity relationships, and PRR targeting beyond TLR2.

#### 5.6.2. Species- and Strain-Specific Effects

Experimental studies suggest that the immunological role of lactobacilli-derived LPPs may vary considerably in strain and host contexts. In human PBMCs, an LPP-deficient mutant of *L. plantarum* triggered the enhanced production of IL-12, TNF-α, IL-1β, and IL-8, and reduced the levels of IL-10 compared to the WT strain, suggesting that intact LPPs may help dampen host inflammatory responses [250]. In contrast, purified LPPs from *L. plantarum*, *L. casei*, or LGG failed to attenuate flagellin-induced IL-8 secretion in PBMCs, indicating a limited immunosuppressive activity in those settings [251].

Nevertheless, other experimental systems have yielded different outcomes. In human myometrial hTERT-HM cells, LPP-enriched fractions isolated from the CFS of *L. rhamnosus* GR-1 significantly reduced the LPS-induced secretion of IL-6, IL-8, and MCP-1, pointing toward a potential anti-inflammatory role in epithelial or barrier tissue contexts [252]. These apparently divergent findings highlight the species- and tissue-dependent complexity of LPP–immune system interactions. Further studies are needed to clarify whether structural differences among LPPs, such as lipid composition, acylation patterns, or protein domains, are responsible for these distinct immunological outcomes.

### 5.7. DNA and CpG-Rich Oligodeoxynucleotides

#### 5.7.1. Structure and Key Features

Bacterial genomic DNA, particularly from lactobacilli, contained immunologically active unmethylated cytosine–phosphate–guanine (CpG) motifs that were recognized by TLR9 located in the endosomal compartments of immune cells (Figure 8). TLR9 detection initiated MyD88-dependent signaling cascades that led to NF-κB and MAPK activation and influenced both innate and adaptive immunity [94,253]. These CpG-rich sequences were abundant in bacterial DNA and represented key MAMPs capable of modulating immune responses in a context- and sequence-dependent manner.

#### 5.7.2. Species- and Strain-Specific Effects

Probiotic-derived genomic DNA (gDNA) exhibited both immunostimulatory and anti-inflammatory properties. DNA from the VSL#3 probiotic formulation induced IL-6 and IL-12p40 secretion in bone marrow-derived macrophages (BMDMs) by activating NF-κB and JNK pathways and protected mice against DSS-induced colitis through a TLR9-MyD88-dependent mechanism [254]. Similarly, gDNA from LGG reduced inflammatory cytokines in LPS-challenged RAW264.7 macrophages and suppressed p38 and NF-κB activation, especially when co-administrated with SLPs [229].

Defined synthetic oligodeoxynucleotides (ODNs) containing CpG motifs derived from lactobacilli DNA, such as ID35 from LGG, activated DCs and promoted T_h_1-biased immune responses in ovalbumin-challenged mice. These responses were accompanied by reduced T_h_2-type cytokines and a decreased IgE production, indicating anti-allergic activity [255]. CpG-ODNs also synergized with TLR4 ligands to amplify IL-12 and TNF-α expression, while increasing TLR9 mRNA and p65 phosphorylation [230]. Characteristic motifs such as 3′-GTCGTT-5′ and 3′-PuPuCGPyPy-5′ were detected in *L. plantarum*, *L. rhamnosus*, *L. casei*, and *L. delbrueckii* [256].

Not all CpG motifs triggered immune activation; some acted suppressively: an example was ODN 7F, a CpG-rich sequence from *L. casei* DNA, which activated TLR9 but reduced pro-inflammatory mediators including MIP-2, iNOS, and COX2 in THP-1 macrophage-like cells, and was confirmed against DSS-induced colitis in mice [257]. Immunosuppressive ODNs from *L. paracasei* reduced IL-6, TNF-α, and IL-12p40 in DCs and promoted T_reg_ differentiation in vivo, contributing to mucosal tolerance [258]. Similar anti-inflammatory effects were reported for ODNs derived from LGG in DSS-induced colitis in mice models [259].

The immunological effects of CpG-rich DNA were species- and host-dependent. DNA from *L. rhamnosus* and *L. fermentum* decreased TNF-α secretion in endotoxin-stimulated RAW264.7 cells in a dose-dependent way [260]. However, the same CpG-rich sequences could exert divergent outcomes depending on the host immune context: DNA from *L. plantarum* reduced NF-κB and STAT1 signaling in colonic biopsies from ulcerative colitis patients, while in Crohn’s disease biopsies it promoted IL-17 expression, indicating a differential recognition or downstream signaling by inflamed tissue [261].

Lactobacilli-derived DNA and CpG-rich ODNs represented potent immunomodulators capable of inducing either inflammatory or regulatory immune responses through TLR9 signaling [257]. Sequence-specific and species-dependent effects highlighted the need for precision in characterizing functional motifs and host–pathogen interactions. Elucidating the structural determinants of immunostimulatory versus suppressive CpG-ODNs remains essential for their therapeutic exploitation in inflammatory and allergic diseases.

### 5.8. Membrane Vesicles (MVs)

#### 5.8.1. Structure and Key Features

MVs constitute spherical, bilayered nanostructures (20–400 nm) naturally released by several lactobacilli [262] (Figure 9). These vesicles enclose cytoplasmic and membrane-derived components, including nucleic acids, proteins, LTA, and PGN fragments, which are implicated in host–microbiome immune interactions [40,263,264]. The internalization of MVs by epithelial and immune cells is required to initiate most immunological effects, as demonstrated in both in vitro and in vivo settings [45,265].

#### 5.8.2. Species- and Strain-Specific Effects

MVs derived from *L. sakei* and *L. plantarum* stimulated IL-6 and IgA production in Peyer’s patches cells via the activation of DCs [266,267]. These responses were shown to be dependent on TLR2 engagement [268]. MVs from *L. plantarum* JCM8341 further activated NF-κB signaling and induced IL-1β, IL-6, IL-10, IFN-γ, and IL-12 in RAW264.7 macrophages, promoting the T_h_1-skewing of naïve T-cells [267]. In HT-29 epithelial cells, the same MVs suppressed LPS-induced IL-8 expression in a dose-dependent manner, while in DSS-induced colitis models they reduced in a concentration-dependent manner. These MVs also alleviated DSS-induced colitis models; they reduced weight loss and tissue damage. These effects were partially mediated by MV-associated small RNAs capable of inhibiting TLR4-dependent pathways [45].

MVs from *L. rhamnosus* JB-1 induced IL-10 secretion in BMDCs and suppressed TNF-α-induced IL-8 expression in intestinal epithelial cells, likely through their LTA content [269]. Vesicles from *L. reuteri* enhanced IL-6 and IL-10 production in resting PBMCs and downregulated IFN-γ and IL-17 in *S. aureus*-stimulated PBMCs [40]. In related experiments, only *L. reuteri*-derived but not LGG-derived MVs attenuated TNF-α and IFN-γ expression, possibly due to strain-specific enzymatic cargo, including 5′-nucleotidase [264].

The immunological effects of MVs were highly variable among strains. *L. reuteri*-derived MVs suppressed intestinal inflammation and oxidative stress in LPS-treated broilers, inhibited pro-inflammatory cytokines (TNF-α, IL-6, IL-7), and enhanced regulatory mediators such as IL-10 and TGF-β. However, MVs downregulated IFN-γ and IL-17 expression in co-cultured splenic lymphocytes while promoting CD25^+^ and CTLS-4, facilitating T_reg_ polarization. Vesicle-associated nucleic acids and membrane proteins were considered critical mediators of these effects [263].

Strain specificity also depended on environmental factors. For instance, *L. casei* MVs produced under agitation and *L. plantarum* MVs generated at a low pH induced greater IL-10 and reduced TNF-α in THP-1 macrophages compared to standard conditions, despite lower vesicle yields [270]. This observation suggested that vesicle bioactivity was sensitive to both genetic and environmental inputs.

All these studies point to different factors responsible for the immunomodulatory effects; the structure complexity of the lactobacilli MVs, as a matter of fact, allows us to focus our attention on multiple MAMPs and molecules, exposed on the membrane surface or localized inside the vesicles, which all probably have a role in influencing the host immune system. Also, we can assert that MV properties are strongly species- and strain-dependent, but also are influenced by the culture conditions the strains have been exposed to.

### 5.9. Immunobiotic-like Particles (IBLPs)

IBLPs, also referred to as bacterial-like particles (BLPs), consist of hollow PGN-based scaffolds mimicking MAMP-rich vesicles. These particles demonstrate a robust immunostimulatory activity in vivo [271]. *L. rhamnosus*-derived IBLPs, when used as a vaccine adjuvant, enhanced Rotavirus-specific IgA and IgG responses in mice and increased the production of TNF-α, IFN-γ, and IL-4 in mucosal and systemic compartments [272]. Other formulations, such as BLP23017, significantly boosted IL-1β, IL-6, IL-12, TNF-α, and IFN-γ secretion in macrophages and elevated mucosal sIgA and serum antibody levels in immunized animals against *Clostridium perfringens* [273].

Both membrane vesicles and IBLPs represent complex, multifactorial delivery system for MAMPs. Their immunomodulatory activity depends on molecular cargo, surface-exposed ligands, and host–cell recognition mechanisms. As strain-specific and environmentally responsive platforms, MVs and IBLPs show strong potential for immunotherapeutic and adjuvant applications, although further mechanistic studies are needed to elucidate their receptor-binding profiles and optimize their functional design.

## 6. How Do Lactobacillaceae MAMPs Differ from Those of Other Bacteria?

Many Gram-positive MAMPs converge on TLR2/NOD2, yet lactobacilli display recurring chemo-structural signatures (e.g., LTA D-alanylation status, specific EPS chemotypes, S-layer/pilus architectures) that bias PRR usage, kinetics, and functional outputs distinct from other commensals or pathogens. The table below (Table 1) highlights reproducible differences with functional implications.

Together, these features support a “lactobacilli MAMP signature”: tunable LTA acylation/D-alanylation, EPS chemotypes with barrier-repair functions, S-layer/DC-SIGN engagement, and sortase-piliated adhesion. While TLR2/NOD2 convergence is shared across Gram-positives, these structural/topological traits bias lactobacillus outputs toward epithelial protection, restrained NF-κB, and regulatory T-cell support more consistently than in many other commensals or pathogens.

## 7. Main Challenges in Understanding Immunomodulation by Lactobacilli

Despite substantial advances in the comprehension of how discrete MAMPs such as PGN, LTA, and EPS engage host immune mechanisms, several critical challenges continue to limit our full understanding of lactobacilli-derived immunomodulation.

A first major limitation resides in the discrepancy between in vitro and in vivo models. While in vitro systems enable a high-resolution analysis of cellular pathways, they often fail to recapitulate the complexity of host physiology, including microbial community interactions, mucosal architecture, and immune cell crosstalk. Conversely, animal models, though more physiologically relevant, exhibit an interspecies variability in immune signaling and microbiome composition, complicating extrapolation to human systems [283,284].

Another key issue lies in the strain-specific nature of MAMP bioactivity. Closely related strains can exhibit markedly different immunomodulatory profiles due to the subtle genomic or regulatory variations affecting MAMP abundance, structure, or release; this high degree of functional heterogeneity presents a challenge for reproducibility and clinical applications [285].

Furthermore, most studies have characterized complex molecular landscapes in which multiple MAMPs act simultaneously. These combinations may generate additive, synergistic, or even antagonistic effects on immune signaling, a dimension that remains largely unexplored and difficult to prove experimentally [286].

Host-specific factors further complicate interpretation. Immune responsiveness to bacterial signals is shaped by host genetics, existing microbiota composition, diet, age, and disease state [287]. Interindividual variability may obscure the generalizability of MAMP-based interventions unless studies account for such biological contexts.

Lastly, regulatory and technological barriers limit the translational potential of MAMPs. Issues surrounding the purity, reproducibility, and scalability of MAMP preparations, as well as the definition of pharmacologically active doses, must be rigorously addressed to meet clinical-grade manufacturing and safety standards.

To overcome these limits, future studies should integrate high-throughput -omics approaches, ex vivo immune profiling, and personalized host–microbiome interaction models. Such integrative strategies are essential to fully harness the therapeutic potential of lactobacilli-derived MAMPs.

## 8. Discussion

Lactobacilli are among the most frequently consumed probiotics, thanks to their beneficial effects on human health. For instance, fermented dairy products containing live and active cultures have been associated with a reduced risk of type II diabetes, better glycemic control [288], reduced weight gain [289], and lower circulating triglyceride levels, systolic blood pressure, and insulin resistance [290]. Various lactobacilli species also appear to play important roles in both the treatment and prevention of antibiotic-associated diarrhea [291], as well as in reducing the incidence or severity of gastrointestinal infections by *Salmonella* spp. [292,293], *Clostridium difficile* [291,294], and *H. pylori* infection [295,296]. Moreover, current research is exploring extra-intestinal applications of lactobacilli for neuronal autoimmune and neurodegenerative diseases, including multiple sclerosis [297], Alzheimer’s disease [298,299], Parkinson’s disease [300], and neurodevelopmental disorders such as ADHD [301].

One of the key reasons for this broad therapeutic potential is that lactobacilli possess various MAMPs. These MAMPs can be recognized by PRRs such as TLRs and NLRs, triggering complex signaling pathways that modulate cellular behavior at the transcriptional level and thus influence immune responses. The net effect can be either pro-inflammatory or anti-inflammatory, depending on the lactobacilli species or strain [123,141,163], the specific MAMP in question, the target cell type, and the activation status of that cell.

A clear example is seen in macrophages: pro-inflammatory (M1) macrophages are typically induced by Th1 cytokines (e.g., IFN-γ, TNF-α) or bacterial LPS and secrete high levels of pro-inflammatory cytokines, whereas anti-inflammatory (M2) macrophages, stimulated by Th2 cytokines, produce cytokines that dampen inflammation [302,303]. Many in vitro studies demonstrate that lactobacilli MAMPs can induce pro-inflammatory effects in resting (unstimulated) macrophages [193], while in LPS-primed macrophages they actually suppress excessive pro-inflammatory cytokine production [199]. The capacity of lactobacilli and their MAMPs to shape immune responses makes them appealing for the prevention and treatment of numerous immune-related disorders. For IBDs, which include CD and UC, lactobacilli have displayed protective activities by inducing mucin-2 and occludin, increasing anti-inflammatory mediators, and reducing IL-6, IL-1β, and TNF-α levels, along with diminishing the presence of CD25^+^ T-cells in inflamed tissues [38,304,305]. Polysaccharide–peptidoglycan complexes from *L. casei* Shirota significantly inhibited IL-6 production in LPS-stimulated colonic lamina propria mononuclear cells, RAW264.7 macrophages, and PBMCs from UC patients, leading to milder symptoms in ileitis mouse models [139]. Peptidoglycan from *L. salivarius* Ls33 was shown to protect BALB/c mice against colitis through NOD2 receptor activation, resulting in elevated colonic IL-10 levels and increased T_reg_ cells in the mesenteric lymph nodes, even with oral administration [141]. LTA has also been studied for its anti-inflammatory effects in colitis models; numerous findings suggest its potential use in alternative IBD therapies [153,161,306]. In addition, EVs derived from lactobacilli can ameliorate colonic inflammation in murine models. For example, EVs from *L. plantarum* Q7 reduced pro-inflammatory cytokines in the colon and serum of DSS-induced colitis mice, potentially through modulating the TLR4/MyD88/NF-κB pathway [307]. Beyond the gut, lactobacilli-derived MAMPs have been examined for their effects on other inflammatory conditions. LTA purified from *L. plantarum* effectively suppressed atherosclerosis, reducing monocyte/macrophage infiltration and the pro-inflammatory response in the aortic sinus of mice, thus mitigating plaque degradation [150]. LTA from LGG showed radioprotective effects in the mouse intestine, acting via TLR2- and COX2-dependent mechanisms in mesenchymal stem cells and promoting epithelial cell migration—an effect that may hold promise for mitigating radiotherapy-induced damage [308]. Moreover, purified EPSs from *L. rhamnosus* KL37 demonstrated immunosuppressive properties in a collagen-induced arthritis model, decreasing serum autoantibody levels and ameliorating disease symptoms by modulating T-cell-dependent humoral responses [309]. Allergy treatment is another area where lactobacilli MAMPs display significant potential. *L. rhamnosus* GG-derived immunostimulatory oligodeoxynucleotides and *L. acidophilus* PGN have both been shown to reduce IgE levels in mouse models [255]. Likewise, EPSs from *L. paracasei* were able to reduce IgE and IL-4 levels in serum and ear tissue in a murine model of contact dermatitis, improving dermal health [310]. Furthermore, some MAMPs from lactobacilli enhance antitumor immunity. An EPS from lactobacilli strains has been reported to increase CCR6+ CD8^+^ T-cell populations in Peyer’s patches, supporting their migration to CCL20-expressing tumor sites and thereby boosting the effects of immune checkpoint blockades (e.g., anti-CTLA-4 mAb) [311]. Similarly, EVs from LGG ATCC 53103 synergized with anti-PD-1 immunotherapy, fostering a higher infiltration of MHC II+ dendritic cells, CD8^+^, and CD4^+^ T-cells into the tumor [312]. The oral administration of *L. reuteri* also promoted antitumor immunity, enhancing responsiveness to checkpoint inhibitors via the production of indole-3-aldehyde in tumor tissue, which supported stronger CD8^+^ T-cell activity [313]. Ongoing research covers an even broader spectrum, including diabetes [314,315], protection against ethanol-induced liver damage [316], and neurodegenerative pathologies [317]. Despite these encouraging data, the precise mechanisms underlying lactobacilli-mediated immunomodulation are not fully elucidated, and many studies still rely on whole-cell effects. Nevertheless, the mounting evidence that isolated MAMPs can regulate immunity in a myriad of contexts has spurred interest in developing them as “postbiotic” solutions, i.e., non-viable microbial products with retained bioactivity. Despite the growing body of preclinical evidence, clinical validation of these effects remains limited. To date, only one clinical trial has explored the therapeutic efficacy of a postbiotic MAMP preparation (derived from *Streptococcus thermophilus* rather than lactobacilli), highlighting the early stage of translation in this field [318]. As their individual and synergistic roles become better understood, lactobacilli-derived MAMPs are emerging as versatile tools for the management of inflammatory and immune-related conditions.

## 9. Conclusions

The growing body of evidence on lactobacilli-derived MAMPs underscores their essential immunomodulatory role. Acting through TLR, NLR, and other signaling pathways, these molecules can drive pro- or anti-inflammatory responses and often remain active in non-viable (postbiotic) formats, expanding their therapeutic scope even for immunocompromised populations.

In this review, it has been shown how specific MAMPs—such as PNG, LTA, EPS, SLPs, lipoproteins, and CpG ODNs—can strengthen epithelial barriers, promote T_regs_, mitigate excess inflammation, or boost pathogen defense. Beyond gut-focused benefits, these immunomodulatory activities extend into areas such as allergy management, autoimmune support, and tumor immunotherapy, indicating a wide range of clinical potential.

Looking ahead, several hurdles remain. Bridging in vitro–in vivo data, clarifying strain-dependent MAMP differences, and optimizing postbiotic formulations are all essential. Still, harnessing lactobacilli MAMPs as targeted immunomodulators remains a promising approach, aligning traditional probiotic usage with next-generation therapies for inflammatory, infectious, and neoplastic diseases.

Looking ahead, the following developments can be anticipated: (a) Formulating stable postbiotic products enriched for specific MAMPs: Methods to isolate and concentrate LTA, EPS, or specialized vesicles are crucial for scaling these therapies. Stabilization protocols, freeze-drying technology, and microencapsulation may enhance shelf-life and targeted delivery. (b) Conducting more human clinical trials: While a solid base of animal data exists, well-controlled human studies are needed to confirm the optimal dosing, safety in immunocompromised subsets, and efficacy in conditions from IBD to allergies and even cancer. (c) Employing advanced “omics” technologies: Genomics, transcriptomics, proteomics, and metabolomics can pinpoint host–MAMP interactions at the molecular level. This approach helps unravel how structural variations in MAMPs (e.g., differences in glycosylation or acylation) translate into distinct immune outcomes, and how individual host factors modulate responses.

Overall, harnessing lactobacilli MAMPs for targeted immunomodulation is a promising and rapidly growing field with enormous therapeutic potential. As the mechanistic underpinnings become clearer, more precise and potent postbiotic interventions are likely to emerge, bridging the gap between traditional probiotic usage and next-generation immunotherapies.

## Figures and Tables

**Figure 1 biomolecules-15-01609-f001:**
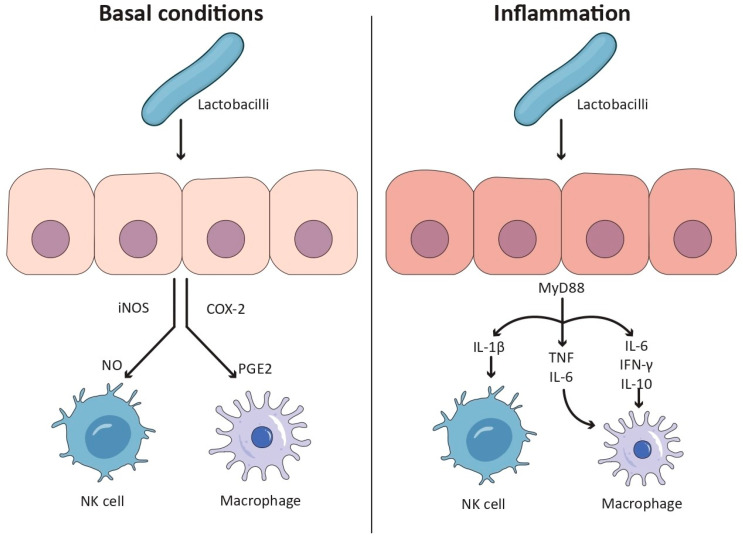
Immunomodulation by lactobacilli under basal conditions and during inflammation. **Left (basal conditions):** tonic epithelial sensing leads to modest iNOS and COX2 induction, generating NO and PGE_2_ that support controlled activity of NK cells and macrophages, thereby maintaining mucosal homeostasis. **Right (inflammation):** in an inflamed context, TLR signaling via MyD88 enhances epithelial release of IL-1β, TNF, and IL-6, reshaping innate responses; macrophages produce IL-6, IFN-γ, and IL-10, integrating pro-inflammatory and regulatory cues. Effects are strain-specific and condition-dependent, varying with the host’s physiological and immunological state.

**Figure 2 biomolecules-15-01609-f002:**
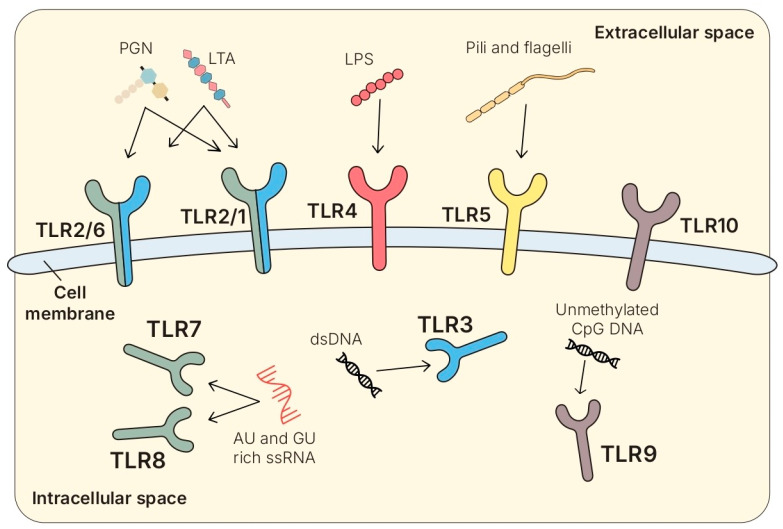
Localization and ligand specificity of human Toll-like receptors (TLRs). A comprehensive overview of TLRs expressed at the cell membrane (TLR1, TLR2, TLR4, TLR5, TLR6, TLR10) and in intracellular compartments (TLR3, TLR7, TLR8, TLR9). The figure illustrates specific microbial ligands for each TLR, such as LPS (TLR4), flagellin (TLR5), lipoproteins, peptidoglycan, and LTA (TLR2/1, 2/6), unmethylated CpG DNA (TLR9), and AU/GU-rich single-stranded RNA (TLR7/8). This spatial segregation enables the coordinated detection of extracellular versus intracellular microbial components and fine-tunes the innate immune response.

**Figure 3 biomolecules-15-01609-f003:**
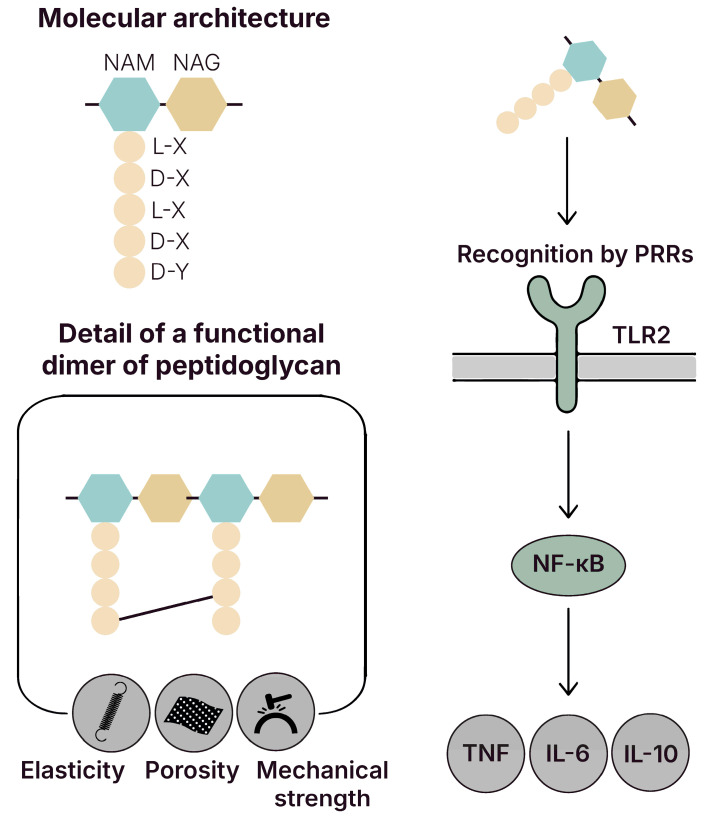
Molecular organization and immunological relevance of bacterial peptidoglycan. The figure illustrates the molecular structure of peptidoglycan (PGN), a major component of Gram-positive bacterial cell walls, composed of repeating disaccharide units of N-acetylmuramic acid (NAM) and N-acetylglucosamine (NAG) cross-linked by peptide chains (L-X, D-X, L-X, D-X, D-Y; this substituent could also be a D-Lactate). These peptide bridges confer mechanical strength, elasticity, and porosity to the bacterial envelope. PGN is sensed by pattern recognition receptors (PRRs), particularly Toll-like receptor 2 (TLR2), triggering activation of the NF-κB signaling pathway and subsequent induction of pro-inflammatory cytokines such as TNF, IL-6, and the regulatory cytokine IL-10. This interaction plays a critical role in the host’s ability to detect and respond to microbial invasion.

**Figure 4 biomolecules-15-01609-f004:**
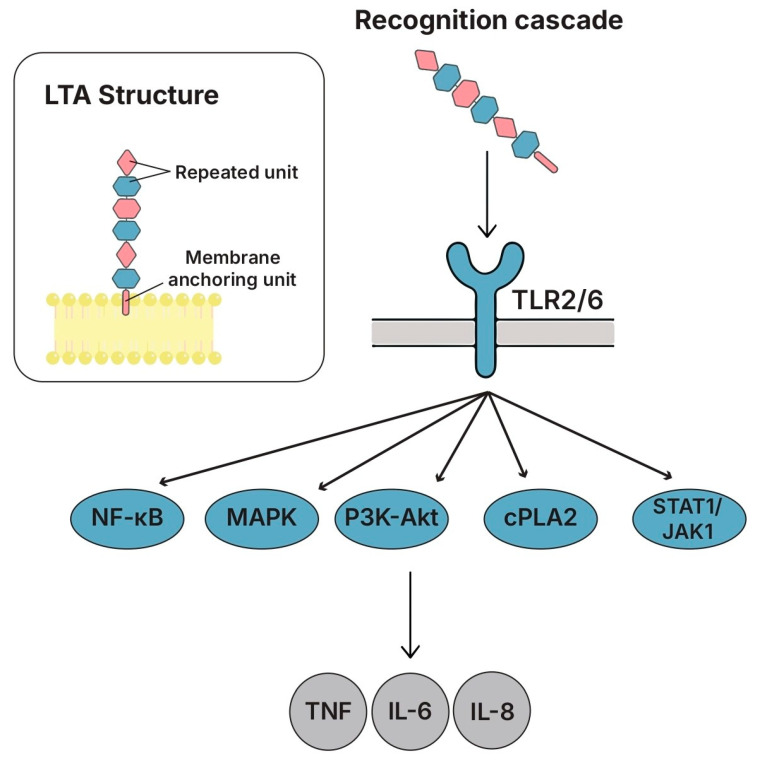
Structural features of lipoteichoic acid (LTA) and downstream immune activation pathways. The figure presents a hypothetical molecular structure of LTA, a glycolipid anchored in the cytoplasmic membrane of Gram-positive bacteria and composed of repeated units. Recognition of LTA by TLR2/6 heterodimers at the host cell surface initiates several signaling cascades, including NF-κB, MAPK, PI3K-Akt, cPLA2, and JAK1/STAT1. These cascades result in transcriptional regulation of inflammatory genes encoding TNF, IL-6, and IL-8, directing an early innate immune response to Gram-positive bacterial infection. LTA thus serves as a MAMP in the modulation of host–pathogen interactions.

**Figure 5 biomolecules-15-01609-f005:**
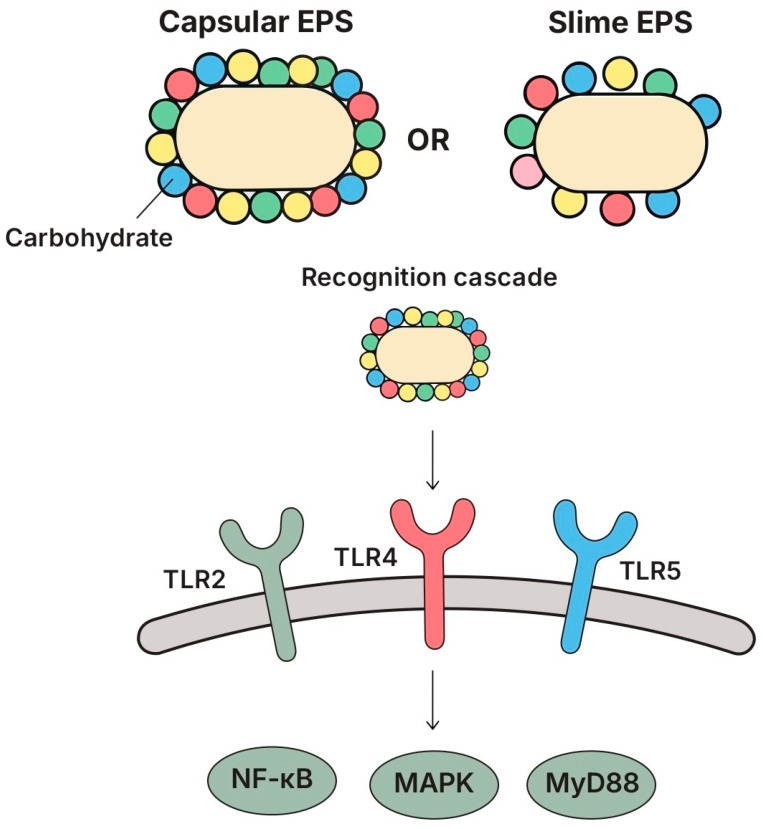
Classification of bacterial exopolysaccharides (EPSs) and their immunostimulatory potential. This scheme distinguishes two main structural types of bacterial EPSs: capsular EPSs, which form rigid external layers, and slime EPSs, which are more loosely associated with the bacterial surface. Both types are primarily composed of complex carbohydrate chains and can be recognized by host PRRs, including TLR2, TLR4, and TLR5. Upon recognition, signaling pathways involving MyD88, MAPK, and NF-κB are activated, culminating in pro-inflammatory cytokine production. EPS molecules not only provide protection to bacteria from host defenses but are also important in immune system modulation and biofilm formation.

**Figure 6 biomolecules-15-01609-f006:**
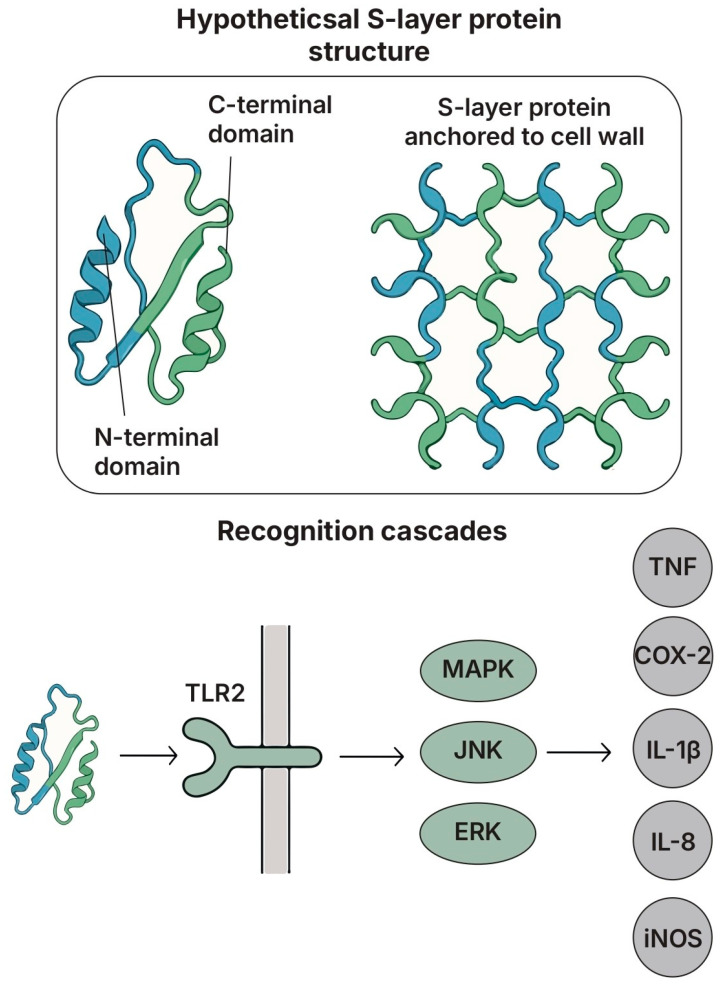
S-layer protein structure and innate immune activation. The figure proposes a model for the structure and immunological function of S-layer proteins, which are para-crystalline arrays found on the surface of many Gram-positive bacteria. These proteins exhibit a modular architecture, with conserved N- and C-terminal domains, and are anchored to the cell wall. Recognition of S-layer proteins by TLR2 on host immune cells leads to activation of Mitogen-Activated Protein Kinase (MAPK) cascades, including ERK and JNK pathways, and promotes transcription of genes involved in the inflammatory response, such as iNOS, COX2, IL-1β, IL-8, and TNF. S-layer proteins contribute to bacterial adhesion, immune evasion, and modulation of host signaling networks.

**Figure 7 biomolecules-15-01609-f007:**
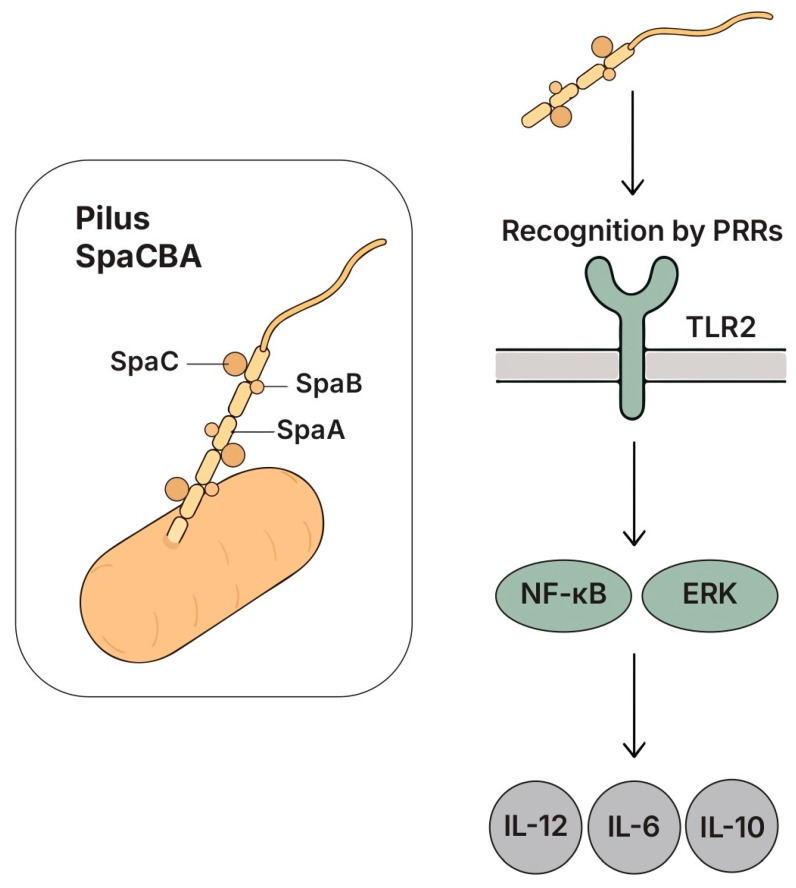
Structure and immunological interaction of the SpaCBA pilus in Gram-positive bacteria. The SpaCBA pilus is a well-characterized sortase-dependent pilus system comprising SpaA (major shaft subunit), SpaB (basal subunit), and SpaC (adhesive tip subunit). This figure highlights the hierarchical organization of pilin subunits and their recognition by TLR2 on host immune cells. SpaCBA engagement triggers the activation of NF-κB and ERK pathways, resulting in the induction of both pro-inflammatory (IL-6, IL-12) and anti-inflammatory (IL-10) cytokines. These pili facilitate adhesion to mucosal surfaces and contribute to bacterial persistence while modulating mucosal immune responses.

**Figure 8 biomolecules-15-01609-f008:**
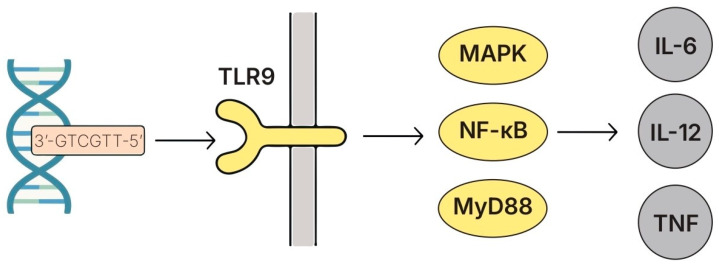
TLR9-mediated recognition of bacterial DNA and resulting immune activation. Bacterial genomic DNA, characterized by unmethylated CpG motifs such as 3′-GTCGTT-5′, is detected by TLR9 in endosomal compartments of immune cells. This figure outlines the molecular events following CpG recognition, which include recruitment of the adaptor protein MyD88, activation of NF-κB and MAPK signaling pathways, and the induction of cytokines such as IL-6, IL-12, and TNF. These responses are essential for initiating adaptive immune responses against intracellular bacterial pathogens and for driving T_h_1 polarization.

**Figure 9 biomolecules-15-01609-f009:**
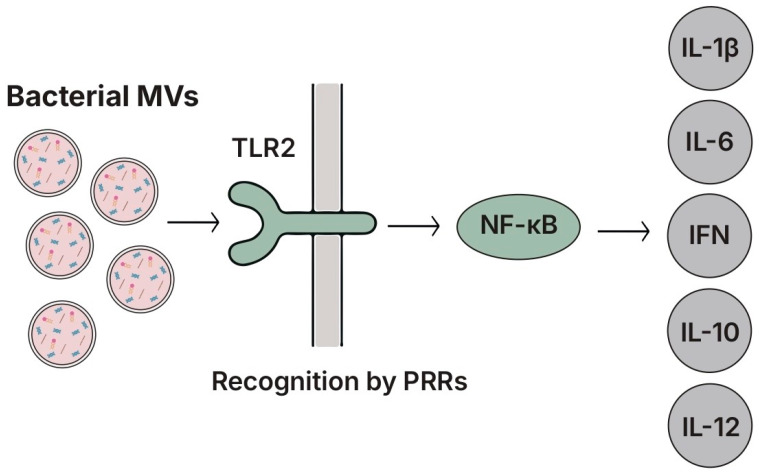
Bacterial membrane vesicles (MVs) as immunomodulatory structures. Depiction of bacterial MVs, which are spherical structures composed of lipid bilayers and enriched in MAMPs. MVs are recognized by host TLR2 and activate NF-κB signaling, leading to the production of interferons and cytokines such as IL-6, IL-1β, IL-10, and IL-12. These vesicles facilitate inter-kingdom communication, contribute to immune modulation, and may act as delivery systems for bioactive compounds during host–microbe interactions.

**Table 1 biomolecules-15-01609-t001:** Comparative features of lactobacilli’s MAMPs versus other genera.

MAMP	Lactobacilli: Recurrent Feature	Comparison with Other Genera and Functional Implication	Refs.
Peptidoglycan	Muropeptide patterns and cell-wall modifications (e.g., O-acetylation via OatA, amidation) tune NOD1/NOD2 sensing and lysozyme susceptibility; in several lactobacilli strains, PGN/fragments tend to dampen IL-12 and NF-κB outputs, favoring regulatory/barrier-supportive profiles.	In pathogens, PGN modifications often promote lysozyme evasion and strong NOD/TLR priming, and the outcome is a pro-inflammatory bias.	[24,118,274,275,276]
Lipoteichoic acid	The degree of D-alanine substitution (operon dlt) acts as a “rheostat”: reduced D-alanylation attenuates TLR2 signaling and inflammation. Lactobacilli LTA can increase IL-10 and reduce MAPK/NF-κB in epithelial/monocytic models.	In many Gram-positive pathogens, highly D-alanylated LTA drives robust TLR2-dependent activation and pro-inflammatory cytokines.	[154,277]
Exopolysaccharides	Frequently acidic/branched, high-MW polymers that modulate TLR2/TLR4–MAPK/STAT pathways, lower pro-inflammatory cytokines, and strengthen tight junctions (barrier-protective profile).	Pathogen EPS/capsules favor adhesion/biofilm and immune evasion; outputs are often pro-inflammatory or broadly suppressive without barrier-repair features.	[25,278,279]
S-layer	S-layer of *L. acidophilus* (SlpA) is a DC-SIGN ligand and cooperates with TLR2 to reshape DC functions and T-cell output (regulatory bias). Recent work reinforces lectin-PRR (DC-SIGN) engagement and immunoregulation in Lactobacilli.	S-layers are not ubiquitous among commensals; in pathogens, S-layers are often linked to innate stimulation (via TLR/CLR) and/or evasion. A DC-SIGN-centric, tolerogenic signature is more typical of lactobacilli.	[221,280,281]
Pili	SpaCBA pili of *L. rhamnosus* GG mediate high-affinity mucus adhesion and can modulate epithelial/DC cytokines (e.g., IL-8); mutant/heterologous systems reveal a contribution of pili to non-pathogenic DC/IEC crosstalk.	In pathogens (e.g., pneumococcus, streptococcus), pili act as pro-inflammatory MAMPs (often TLR2-dependent), boosting IL-8/TNF-α and virulence.	[238,239,282]

## Data Availability

Data is contained within the article and Appendix A.

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
