# Peer review of "Lactobacilli-Derived Microbe-Associated Molecular Patterns (MAMPs) in Host Immune Modulation"

_biomolecules, 2025, doi:10.3390/biom15111609_

Round 1
Reviewer 1 Report
Comments and Suggestions for Authors
This manuscript, entitled "Lactobacilli-Derived Microbe-Associated Molecular Patterns (MAMPs) in Host Immune Modulation and Therapy," presents a comprehensive review of the immunomodulatory properties of non-viable Lactobacillus components, including PGN, LTA, EPS, and SLPs. This manuscript is systematically organized, progressing from live probiotics to specific MAMPs and their target pathways. However, it has several critical and major deficiencies that preclude its acceptance in its current form.
Major comments:
- Figures 3, 4, 5, 6, 7, 8, and 9 follow the same simplistic template: [MAMP Structure] -> [PRR Recognition] -> [Signaling Pathway] -> [Cytokine Output]. Although visually distinct, this repetition becomes conceptually redundant after the initial instance, and provides minimal informational value.
- A significant omission is the lack of mechanistic discussions regarding the context-dependent effects of MAMPs. The review notes that MAMPs can be pro-inflammatory in resting states but anti-inflammatory in inflamed states, yet it fails to critically analyze the reasons for this immunological switch. A more thorough examination of the host-side mechanisms (e.g., differential PRR expression, pathway pre-activation, or epigenetic changes) that dictate these divergent outcomes is necessary.
- This review lacks a comparative analysis, rendering it unclear whether the described immunomodulatory effects are specific to lactobacilli. The manuscript does not discuss how Lactobacilli MAMPs (e.g., PGN or LTA structure) differ from those of other bacterial genera (such as other commensals or pathogens). Without this comparison, the specificity and uniqueness of Lactobacilli-derived MAMPs have not yet been established.
- The title of the manuscript prominently incorporates the term "Therapy," yet the review primarily focuses on preclinical findings and their potential therapeutic implications, particularly in IBD, allergy, and cancer models. There is a conspicuous lack of critical discussion regarding established clinical therapies, human clinical trials, and the translational challenges associated with the development of these MAMPs into actual therapeutic agents. Consequently, the title establishes the expectation that the manuscript will not be fulfilled.
Minor comments:
- The captions for Figure 2 and 3 are identical. Page 1, line 18: " Tumour necrosis factor-alfa ". Change "alfa" to "alpha".
Author Response
Major comments:
- Figures 3, 4, 5, 6, 7, 8, and 9 follow the same simplistic template: [MAMP Structure] -> [PRR Recognition] -> [Signaling Pathway] -> [Cytokine Output]. Although visually distinct, this repetition becomes conceptually redundant after the initial instance, and provides minimal informational value.
1.Thank you for this observation. In this narrative review, the figures are intentionally templated as [MAMP structure → PRR recognition → signaling → cytokine output] to enable rapid, side-by-side comparison across different MAMPs and to help readers immediately identify the core, conserved steps while the mechanistic overviews are discussed step-by-step in the corresponding text sections.
- A significant omission is the lack of mechanistic discussions regarding the context-dependent effects of MAMPs. The review notes that MAMPs can be pro-inflammatory in resting states but anti-inflammatory in inflamed states, yet it fails to critically analyze the reasons for this immunological switch. A more thorough examination of the host-side mechanisms (e.g., differential PRR expression, pathway pre-activation, or epigenetic changes) that dictate these divergent outcomes is necessary.
2.We thank the reviewer for this valuable suggestion. We fully embraced the request and substantially expanded the mechanistic discussion of the context-dependent effects of MAMPs. Specifically, we added a dedicated section (Section 3.3) that analyzes, in a focused and critical way, the host-side determinants that gate the “pro- vs anti-inflammatory” switch. To properly contextualize these mechanisms, we increased the number of references which we believe was necessary to support the added analysis. We trust that these revisions address the concern and materially strengthen the manuscript.
- This review lacks comparative analysis, rendering it unclear whether the described immunomodulatory effects are specific to lactobacilli. The manuscript does not discuss how Lactobacilli MAMPs (e.g., PGN or LTA structure) differ from those of other bacterial genera (such as other commensals or pathogens). Without this comparison, the specificity and uniqueness of Lactobacilli-derived MAMPs have not yet been established.
3.We thank the reviewer for this important point. We have added a dedicated comparative section “How do Lactobacillaceae MAMPs differ from those of other bacteria?” (sec. 6) and a concise side-by-side Table 1.
- The title of the manuscript prominently incorporates the term "Therapy," yet the review primarily focuses on preclinical findings and their potential therapeutic implications, particularly in IBD, allergy, and cancer models. There is a conspicuous lack of critical discussion regarding established clinical therapies, human clinical trials, and the translational challenges associated with the development of these MAMPs into actual therapeutic agents. Consequently, the title establishes the expectation that the manuscript will not be fulfilled.
4.Thank you for this thoughtful observation. We have revised the title to better reflect the manuscript’s actual scope. We did not add a dedicated section on human clinical trials or established therapies because, after an extensive literature search, we found insufficient clinical trial evidence to contextualize the topic in a rigorous and balanced manner.
In light of this, we opted to align the title with the current evidence base and to focus the review on in vitro and preclinical in vivo studies and their potential therapeutic implications. We believe this change sets appropriate expectations for readers while preserving accuracy and clarity.
Minor comments:
1.The captions for Figure 2 and 3 are identical. Page 1, line 18: " Tumour necrosis factor-alfa ". Change "alfa" to "alpha".
1.Thank you for catching these issues. We have (i) revised the captions so that Figure 2 and Figure 3 now have distinct, content-specific descriptions, and (ii) corrected the typo on Page 1, line 18 from “Tumour necrosis factor-alfa” to “Tumour necrosis factor-alpha.”
Reviewer 2 Report
Comments and Suggestions for Authors
I thoroughly enjoyed reading this review article. The authors presented the issues related to the proposed topic in a clear, simple, yet thorough and precise manner. The conclusions are cautious and point to both the broad potential for the use of lactobacilli MAMPs and the vast amount of research still to be conducted.
I have only minor comments to the manuscript:
Please use “lactobacilli” (all small letters) throughout the manuscript (abstract, line 133, 462, 697)
Line 125 microorganisms – particularly lactobacilli – was
3.1.3 and 3.1.4 Administration of live lactobacilli
Line 172 please revise the subtitle or the content of this section, as there are both in vitro and in vivo studies cited
Line 194 “mucin and occludin”
Line 207 “cytotoxic”
Line 220 “immunomodulatory”
Line 421 please provide strain symbol of Lactobacillus acidophilus
Lines 479-480 the sentence seems to be cut. Please correct.
Line 508-510 please reformulate the sentence - L. acidophilus NCK2025 cannot exhibit expression of “MHC-II, CD40, CD86, CD80 and lower cytokine levels (TNF-α, IL-6 and IL-12),…”
Line 510 “In colitis models,…”
Line 514 “Exopolysaccharides“
Lines 580-587 please divide this 7-line sentence.
Line 712 “[220-222].”
Author Response
I have only minor comments to the manuscript:
- Please use “lactobacilli” (all small letters) throughout the manuscript (abstract, line 133, 462, 697)
- Done
- Line 125 microorganisms – particularly lactobacilli – was
- Done
- 1.3 and 3.1.4 Administration of live lactobacilli
- Done
- Line 172 please revise the subtitle or the content of this section, as there are both in vitro and in vivo studies cited
- We thank the reviewer for this helpful observation. After revisiting the relevant paragraphs, we have merged the sections into a single integrated unit that explicitly encompasses both in vitro and in vivo
- Line 194 “mucin and occludin”
- Done
- Line 207 “cytotoxic”
- Done
- Line 220 “immunomodulatory”
- Done
- Line 421 please provide strain symbol of Lactobacillus acidophilus
- Done
- Lines 479-480 the sentence seems to be cut. Please correct.
- Thank you for noting the truncation. The sentence has been corrected to read:
“In L. rhamnosus GG, deprivation of D-Ala residues on LTA yielded a distinct phenotype: in the DSS-induced colitis mouse model, the modified LTA elicited stronger immunomodulatory effects than the wild-type LTA.”
- Thank you for noting the truncation. The sentence has been corrected to read:
- Line 508-510 please reformulate the sentence - acidophilus NCK2025 cannot exhibit expression of “MHC-II, CD40, CD86, CD80 and lower cytokine levels (TNF-α, IL-6 and IL-12),…”
- The requested modification has been implemented. We have reformulated the sentence to clarify that these markers and cytokines refer to host cells and to ensure grammatical parallelism.
- Line 510 “In colitis models,…”
- Done
- Line 514 “Exopolysaccharides“
- Done
- Lines 580-587 please divide this 7-line sentence.
- Thanks fort the comment, we have divided the long sentence into three concise sentences and corrected a minor typographical error.
- Line 712 “[220-222].”
- Done
Reviewer 3 Report
Comments and Suggestions for Authors
Review Report on the Manuscript
“Lactobacilli-Derived Microbe-Associated Molecular Patterns (MAMPs) in Host Immune Modulation and Therapy” Submitted to Biomolecules**
General Comments
The manuscript is well-written, clearly structured, and addresses a relevant and timely topic in the field of biomolecules and immunology. The authors provide a comprehensive overview of Lactobacilli-derived microbe-associated molecular patterns (MAMPs) and their role in immune modulation and therapeutic applications. The review is of potential interest to a wide readership and may serve as a useful reference for researchers working in microbiome-based therapeutics and immunomodulation.
Overall, the study is promising and could be recommended for publication after minor revisions.
Major Comments
- Introduction:
The introduction should provide stronger contextualization of the research problem. Additional recent references should be included to reflect the latest progress in the field. - Discussion:
The discussion is rather descriptive at present. A deeper analytical perspective is needed, especially in terms of highlighting the implications of the findings and comparing them with prior studies. This would significantly strengthen the impact of the manuscript. - Methods and Transparency:
Although the methodology is generally acceptable, some important details remain unclear (e.g., reproducibility, sample handling, specific experimental parameters). These should be clarified for transparency. - Limitations:
A clear statement of the study’s limitations is currently missing. A dedicated paragraph or section is recommended to provide balance and transparency.
Minor Comments
- Language and Style: The manuscript would benefit from careful proofreading to correct minor grammatical issues and improve readability.
- Figure Legends: Legends should be expanded with more descriptive detail for clarity.
- References: There are inconsistencies in formatting that should be corrected for uniformity.
- Results Section: A few clarifications are required to avoid ambiguity and improve precision.
Author Response
Major Comments
- Introduction:
The introduction should provide stronger contextualization of the research problem. Additional recent references should be included to reflect the latest progress in the field.
- Thank you for this suggestion. We respectfully note that the research problem is already contextualized with up-to-date literature. Specifically, among the references cited in the Introduction, approximately 60% are from the last 5 years (2020–2024) and nearly 90% from the last 10 years (2015–2024). In our view, this reflects the latest progress in the field.
The Introduction is designed to frame the problem and delineate the scope, while the subsequent sections of this narrative review examine each aspect in depth. For this reason, details are intentionally developed later rather than step-by-step within the introductory paragraph. We therefore believe the current Introduction is appropriately contextualized and aligned with the structure of the review.
- Discussion:
The discussion is rather descriptive at present. A deeper analytical perspective is needed, especially in terms of highlighting the implications of the findings and comparing them with prior studies. This would significantly strengthen the impact of the manuscript. - Thank you for this observation. We wish to clarify that the manuscript is a narrative review, not an original research article; accordingly, we did not conduct new experiments or generate novel results. The Discussion was intended to contextualize the current state of the art and to outline the future implications of the topic.
- Methods and Transparency:
Although the methodology is generally acceptable, some important details remain unclear (e.g., reproducibility, sample handling, specific experimental parameters). These should be clarified for transparency.
- We thank the Reviewer for the suggestion. However, we respectfully note that the submitted manuscript is a narrative review rather than an original research article, and it is not a systematic review nor a meta-analysis. Consequently, items such as sample handling, experimental reproducibility, or specific experimental parameters are not applicable, since no new experiments or primary data were generated and no quantitative evidence synthesis was performed.
- Limitations:
A clear statement of the study’s limitations is currently missing. A dedicated paragraph or section is recommended to provide balance and transparency. - Thank you for the suggestion. We would like to note that a dedicated Limitations section was already included at the time of submission and remains in the revised manuscript (lines 890–919).
Minor Comments
- Language and Style: The manuscript would benefit from careful proofreading to correct minor grammatical issues and improve readability.
- Thank you for the suggestion. We have carefully re-read the manuscript and implemented a thorough language edit to improve grammar, syntax, and readability.
- Figure Legends: Legends should be expanded with more descriptive detail for clarity.
- Thank you for the suggestion. In this narrative review, the mechanistic steps are explained step-by-step in the corresponding text sections. The figure legends are intentionally concise to synthesize the content of those sections and to help readers immediately identify the key components of each MAMP-related pathway, with the detailed explanations provided in the main text.
- References: There are inconsistencies in formatting that should be corrected for uniformity.
- Thank you for pointing this out. We have conducted a comprehensive reference audit and harmonized the formatting according to the journal’s guidelines
- Results Section: A few clarifications are required to avoid ambiguity and improve precision.
Thank you for the observation. We wish to clarify that this manuscript is a narrative review, and therefore it does not contain a stand-alone Results section as in original research articles.
Round 2
Reviewer 1 Report
Comments and Suggestions for Authors
I appreciate your efforts in addressing the concerns and questions raised in my initial review. Your detailed explanations and clarifications have significantly improved the manuscript. I look forward to seeing it published.